# Reversible Instance Normalization for Accurate Time-Series Forecasting against Distribution Shift

**Taesung Kim**[*]
KAIST AI
zkm1989
@kaist.ac.kr

**Jinhee Kim**[*]
KAIST AI
seharanul17
@kaist.ac.kr

**Yunwon Tae**
VUNO
yunwon.tae
@vuno.co

**Cheonbok Park**
NAVER Corp.
cbok.park
@navercorp.com

**Jang-Ho Choi**
ETRI
janghochoi
@etri.re.kr

**Jaegul Choo**
KAIST AI
jchoo
@kaist.ac.kr

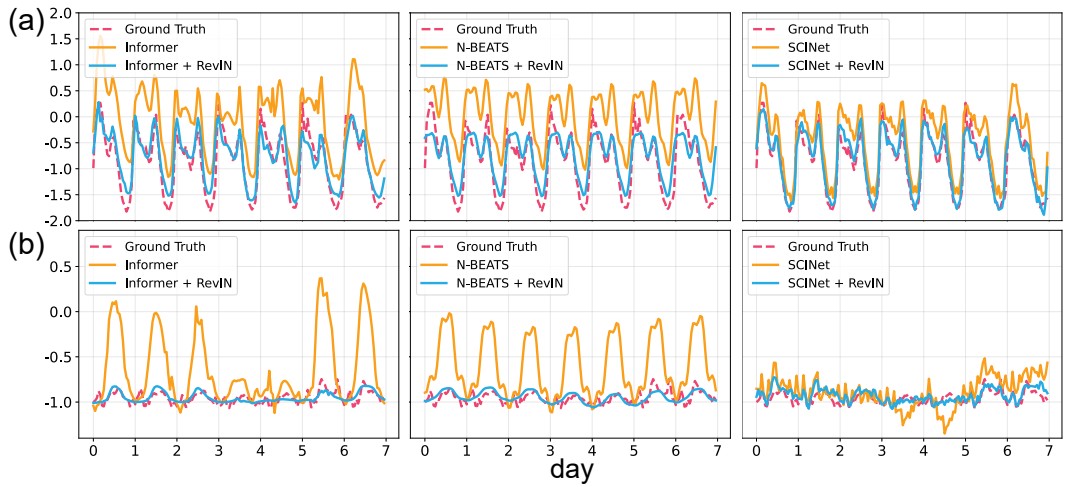

Figure 1: Multivariate time-series forecasting results comparing our method with the state-of-the-art baselines, i.e., Informer (Zhou et al., 2021), N-BEATS (Oreshkin et al., 2020), and SCINet (Liu et al., 2021). The analysis is conducted on electricity consuming load (ECL) dataset, with a prediction length of seven days. The predictions of the baselines are inaccurately (a) shifted and (b) scaled. When adopted to the baselines, our method significantly improves their forecasting performance and better aligns the distribution of the prediction results with the groundtruth values.

## Abstract

Statistical properties such as mean and variance often change over time in time series, i.e., time-series data suffer from a distribution shift problem. This change in temporal distribution is one of the main challenges that prevent accurate time-series forecasting. To address this issue, we propose a simple yet effective normalization method called reversible instance normalization (RevIN), a generally-applicable normalization-and-denormalization method with learnable affine transformation. The proposed method is symmetrically structured to remove and restore the statistical information of a time-series instance, leading to significant performance improvements in time-series forecasting, as shown in Fig. 1. We demonstrate the effectiveness of RevIN via extensive quantitative and qualitative analyses on various real-world datasets, addressing the distribution shift problem.

---

[*]Both authors contributed equally. The order of the first authors was determined by coin flip.

# 1 INTRODUCTION

Time-series forecasting plays a significant role in addressing various daily problems, including health care, economics, and traffic data analyses (Kim et al., 2021a; Ahmadi et al., 2019; Park et al., 2020). Recently, time-series forecasting models have achieved outstanding performance on these problems, overcoming several challenges, such as long-term forecasting (Zhou et al., 2021; Liu et al., 2021) and missing value imputation (Zhang et al., 2021; Kim et al., 2021b). However, the time-series forecasting models often suffer badly from a unique characteristic in time-series data: their statistical properties, e.g., mean and variance, can change over time. This is widely known as the distribution shift problem, and it can yield discrepancies between the distributions of the training and test data of the forecasting models. In time-series forecasting tasks, the training and test data are usually divided from the original data based on a specific point in time. Accordingly, they often hardly overlap, which is a common reason for model performance degradation. Furthermore, the input sequences to the model can have different underlying distributions as well. We can assume that the discrepancy between different input sequences can significantly degrade the model performance.

Under this assumption, if we remove non-stationary information from the input sequences, specifically, the mean and standard deviation of the instances, the discrepancy in the data distributions will be reduced, thereby improving model performance. However, applying such normalization to the model input can cause another problem since it can prevent the model from capturing the original data distribution. It removes non-stationary information that can be important to predict future values in the forecasting task. The model would need to reconstruct the original distribution only using the normalized input, which degrades its forecasting performance due to the inherent limitation. Thus, if we explicitly return the information removed by input normalization back to the model, the model will not have to rebuild the original distribution by itself while keeping the advantage of normalizing the input. To accomplish this, we propose to reverse the normalization applied to the input data in the output layer, i.e., to denormalize the model output using the normalization statistics.

Inspired by this, we propose a simple yet effective normalization-and-denormalization method, **reversible instance normalization (RevIN)**, which first normalizes the input sequences and then denormalizes the model output sequences to solve the time-series forecasting problems against distribution shift. RevIN is symmetrically structured to return the original distribution information to the model output by scaling and shifting the output in the denormalization layer in an amount equivalent to the shifting and scaling of the input data in the normalization layer. To verify the effectiveness of RevIN, we conduct extensive quantitative evaluations using several state-of-the-art time-series forecasting methods as the baselines: Informer (Zhou et al., 2021), N-BEATS (Oreshkin et al., 2020), and SCINet (Liu et al., 2021). We also provide an in-depth analysis of the behavior of the proposed approach, including verification of the assumptions on reversible instance normalization.

RevIN is a flexible, end-to-end trainable layer that can be applied to any arbitrarily chosen layers, effectively suppressing non-stationary information (mean and variance of the instance) in one layer and restoring it in another layer at a virtually symmetric position, e.g., input and output layers. Despite its remarkable performance, there has been no work on generalizing and expanding instance-wise normalization-and-denormalization as a flexibly applicable, trainable layer in the time-series domain. Recently, deep learning-based time-series forecasting approaches, such as Informer (Zhou et al., 2021) and N-BEATS (Oreshkin et al., 2020), have shown outstanding performance in time-series forecasting. However, they have overlooked the importance of normalization, merely using simple global preprocessing of the model input without further exploration and expecting their end-to-end deep learning model to replace the role. Despite the simplicity of our method, there have been no cases of using such techniques in modern deep-learning-based time-series forecasting approaches (Zhou et al., 2021; Liu et al., 2021; Oreshkin et al., 2020). In this sense, we introduce the importance of an appropriate normalization method for deep-learning-based time-series approaches. We propose a carefully designed, deep-learning-friendly module for time-series forecasting by combining the method with the learnable affine transformation, which has been widely accepted in recent deep-learning-based normalization work (Ulyanov et al., 2016).

In summary, our contributions are as follows:

- We propose a simple yet effective normalization-and-denormalization method for time-series, called RevIN, which is symmetrically structured to remove and restore the statisti-

cal information of a time-series instance. The proposed method is generally applicable to arbitrary deep neural networks with negligible cost.

- By adding RevIN to the baseline, we achieve state-of-the-art performance on seven large-scale real-world datasets by a significant margin.

- We conduct extensive evaluations of RevIN using quantitative analysis and qualitative visualizations to verify its effectiveness, addressing the distribution shift problem.

## 2 RELATED WORK

**Time-series forecasting.** Time-series forecasting methods are mainly categorized into three distinct approaches: (1) statistical methods, (2) hybrid methods, and (3) deep learning-based methods. Statistical models are theoretically well guaranteed and have several advantages, including interpretability. As an example of the statistical models, exponential smoothing forecasting (Holt, 2004; Winters, 1960) is a well-established benchmark for predicting future values. To further boost performance, recent work proposed a hybrid model (Smyl, 2020) that incorporates a deep learning module with a statistical model. It achieved better performance than statistical methods in the M4 time-series forecasting competition. The deep learning-based method basically follows the sequence-to-sequence framework to model the time-series forecasting. Initially, deep learning-based models utilized variations of recurrent neural networks (RNNs). However, to overcome the limitation of the limited receptive field, several studies utilized advanced techniques, such as the dilatation and attention module. For instance, SCINet (Liu et al., 2021) and Informer (Zhou et al., 2021) modified the sequence-to-sequence-based model to improve performance for long sequences. However, most previous deep learning-based models are hard to interpret compared to statistical models. Thus, inspired by statistical models, N-BEATS (Oreshkin et al., 2020) designed an interpretable layer for time-series forecasting by encouraging the model to learn trend, seasonality explicitly, and residual components. This model shows superior performance on the M4 competition dataset.

**Distribution shift.** Although there are various models for time-series forecasting, they often suffer from non-stationary time-series, where the data distribution changes over time. Domain adaptation (Tzeng et al., 2017; Ganin et al., 2016; Wang et al., 2018) and domain generalization (Wang et al., 2021; Li et al., 2018; Muandet et al., 2013) are common ways to alleviate the distribution shift. A domain adaptation algorithm attempts to reduce the distribution gap between source and target domains. A domain generalization algorithm only relies on the source domain and hopes to generalize on the target domain. Both domain adaptation and generalization have a common objective, which bridges the gap between source and target distributions. However, defining a domain is not straightforward in non-stationary time series since the data distribution shifts over time. Recently, Du et al. (Du et al., 2021) proposed Adaptive RNNs to handle the distribution shift problems of non-stationary time-series data. It first characterizes the distribution information by splitting the training data into periods. Then, it matches the distributions of the discovered periods to generalize the model. However, unlike Adaptive RNNs, which is costly, RevIN is simple yet effective and model-agnostic. The method can be easily adopted to any deep neural network.

## 3 PROPOSED METHOD

This section proposes reversible instance normalization to alleviate the distribution shift problem in time-series, which is known to cause a substantial discrepancy between the training and test data distributions. Section 3.1 describes the proposed method in detail, and Section 3.2 discusses how our approach mitigates the distribution discrepancy in time-series data.

### 3.1 REVERSIBLE INSTANCE NORMALIZATION

Given a set of input $\mathcal{X} = \{x^{(i)}\}_{i=1}^{N}$ and the corresponding target $\mathcal{Y} = \{y^{(i)}\}_{i=1}^{N}$, we consider a multivariate time-series forecasting task in discrete time, where $N$ denotes the number of sequences. Let $K, T_x$, and $T_y$ denote the number of variables, the input sequence length, and the model prediction length, respectively. Given an input sequence $x^{(i)} \in \mathbb{R}^{K \times T_x}$, we aim to solve the time-series forecasting problem, which is to predict the subsequent values $y^{(i)} \in \mathbb{R}^{K \times T_y}$. In RevIN, the input

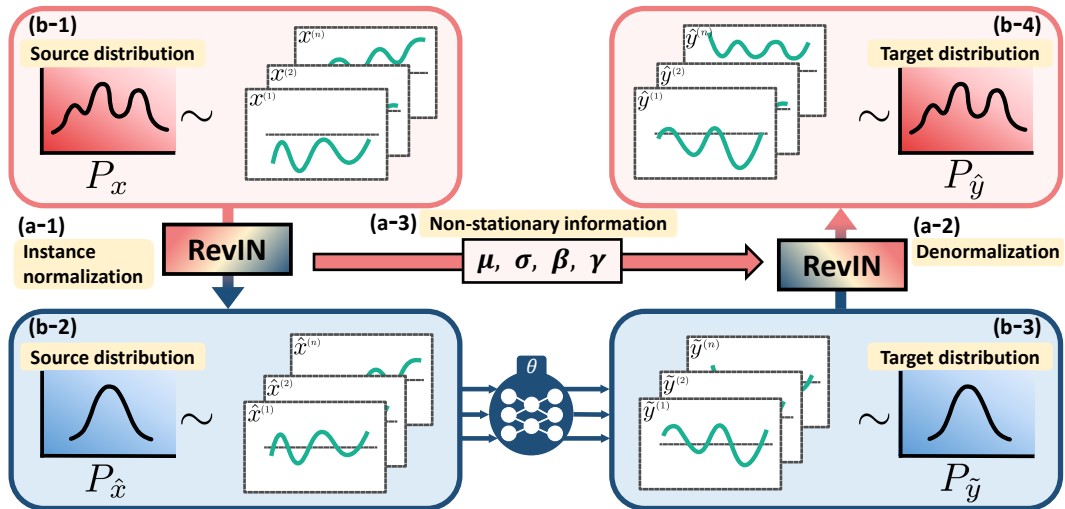

Figure 2: **Overview of the proposed method.** We illustrate an example of a univariate case, where $x^{(i)} \in \mathbb{R}^{1 \times T_x}$; the input data $x^{(i)}$ is actually multivariate (See Section 3.1). In RevIN, the (a-1) instance normalization and (a-2) denormalization are symmetrically structured to remove (a-3) non-stationary information from one layer and restore it on the other layer. Here, RevIN is applied to the input and output layers. The (a-3) non-stationary information includes statistical properties from the input data: mean $\mu$, variance $\sigma^2$, and learnable affine parameters $\gamma, \beta$. The normalization layer transforms the (b-1) original data distribution into a (b-2) mean-centered distribution, where the distribution discrepancy between different instances is reduced. Using $\hat{x}$, the model predicts the future values $\tilde{y}$ following the (b-3) distribution where non-stationary information is eliminated. To restore it (b-4), RevIN reverses the instance normalization in the output layer.

sequence length $T_x$ and the prediction length $T_y$ can be different since the observations are normalized and denormalized across the temporal dimension, as will be explained below. Our proposed method, RevIN, consists of symmetrically structured normalization-and-denormalization layers, as illustrated in Fig. 2. First, we normalize the input data $x^{(i)}$ using its instance-specific mean and standard deviation, which is widely accepted as instance normalization (Ulyanov et al., 2016). The mean and standard deviation are computed for every instance $x_{k \cdot}^{(i)} \in \mathbb{R}^{T_x}$ of the input data (Fig. 2(a-3)) as

$$\mathbb{E}_t[x_{kt}^{(i)}] = \frac{1}{T_x} \sum_{j=1}^{T_x} x_{kj}^{(i)} \qquad \text{and} \qquad \text{Var}[x_{kt}^{(i)}] = \frac{1}{T_x} \sum_{j=1}^{T_x} \left(x_{kj}^{(i)} - \mathbb{E}_t[x_{kt}^{(i)}]\right)^2. \qquad (1)$$

Using these statistics, we normalize the input data $x^{(i)}$ (Fig. 2(a-1)) as

$$\hat{x}_{kt}^{(i)} = \gamma_k \left( \frac{x_{kt}^{(i)} - \mathbb{E}_t[x_{kt}^{(i)}]}{\sqrt{\text{Var}[x_{kt}^{(i)}] + \epsilon}} \right) + \beta_k, \qquad (2)$$

where $\gamma, \beta \in \mathbb{R}^K$ are learnable affine parameter vectors. The normalized sequences can have a more consistent mean and variance, where the non-stationary information is reduced. As a result, the normalization layer allows the model to accurately predict the local dynamics within the sequence while receiving inputs of consistent distributions in terms of the mean and variance.

The model then receives the transformed data $\hat{x}^{(i)}$ as input and forecasts their future values. However, the input data have different statistics than the original distribution, and by observing only the normalized input $\hat{x}^{(i)}$, it is difficult to capture the original distribution of the input $x^{(i)}$. Thus, to make this easier for the model, we explicitly return the non-stationary properties removed from the input data to the model output by reversing the normalization step at a symmetric position, the output layer. A denormalization step can return the model output to the original time-series value as well (Ogasawara et al., 2010). Accordingly, we denormalize the model output $\tilde{y}^{(i)}$ by applying the

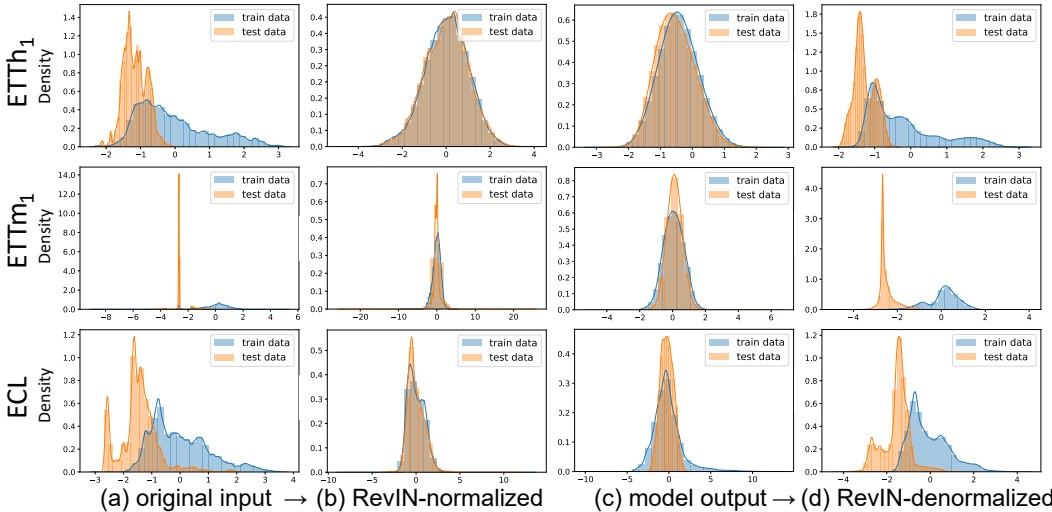

Figure 3: **Effect of RevIN on distribution discrepancy between training and test data.** From left to right columns, we compare the training and test data distributions of a variable on each step of the sequential process in RevIN: (a) the original input $x$, (b) the input $\hat{x}$ normalized by RevIN, (c) the model prediction output $\tilde{y}$, and (d) the output $\hat{y}$ denormalized by RevIN, the final prediction. The analysis is conducted on the ETT and ECL datasets using SCINet (Liu et al., 2021) as the baseline.

reciprocal of the normalization in Eq. 2 (Fig. 2(a-3)) as

$$\hat{y}_{kt}^{(i)} = \sqrt{\mathrm{Var}[x_{kt}^{(i)}] + \epsilon} \cdot \left( \frac{\tilde{y}_{kt}^{(i)} - \beta_k}{\gamma_k} \right) + \mathbb{E}_t[x_{kt}^{(i)}]. \tag{3}$$

The same statistics used in the normalization step in Eq. 2 are used for the scaling and shifting. Now, $\hat{y}^{(i)}$ is the final prediction of the model instead of $\tilde{y}^{(i)}$.

Simply added to virtually symmetric positions in a network, RevIN can effectively alleviate distribution discrepancy in time-series data, as a generally-applicable trainable normalization layer to arbitrary deep neural networks. Indeed, the proposed method is a flexible, end-to-end trainable layer that can be applied to any arbitrarily chosen layers, even to several layers. We verify its effectiveness as a flexible layer by adding it to the intermediate layers in the model in Table 7 in Appendix A.4. Nevertheless, RevIN is most effective when applied to virtually symmetric layers of encoder-decoder structure. In a typical time-series forecasting model, the boundary between the encoder and the decoder is often unclear. Thus, we apply RevIN to the input and output layers of a model as they can be interpreted as an encoder-decoder structure, generating subsequent values, given input data.

## 3.2 EFFECT OF REVERSIBLE INSTANCE NORMALIZATION ON DISTRIBUTION SHIFT

This section verifies that RevIN can alleviate the distribution discrepancy problem by removing non-stationary information in the input layer and then restoring it in the output layer. We analyze the distributions of the training and test data at each step of the proposed approach, as shown in Fig. 3.

When comparing the distribution of training and test data in each example (Fig. 3(a-b)), we can observe that RevIN significantly reduces their discrepancy. To be specific, in the original input (Fig. 3(a)), the training and test data distributions hardly overlap (especially ETTm$_1$), which is caused by the distribution shift problem. Also, each data distribution has multiple peaks (especially the test data of ETTh$_1$ and ECL), implying that sequences in the data might have severe discrepancies in their distributions. However, in the proposed approach, the normalization step transforms each data distribution into mean-centered distributions (Fig. 3(b)). This result supports that the original multimodal distributions (Fig. 3(a)) are caused by discrepancies in distributions between different sequences in the data. Even more, the proposed approach makes training and test data distributions overlapped. This verifies that the normalization step of RevIN can alleviate the distribution shift problem, reducing the distribution discrepancy between training and test data.

Taking the normalized data as the input, the model can retain aligned training and test data distributions in the prediction output (Fig. 3(c)). As expected, these are then returned back to the original distribution by the denormalization step of RevIN (Fig. 3(d)). Without denormalization, the model needs to reconstruct the values that follow the original distributions (Fig. 3(d)) using only the normalized input that follows the transformed distributions where non-stationary information is removed (Fig. 3(b)). Additionally, we hypothesize that the distribution discrepancy will be reduced in the intermediate layers of the model as well, when RevIN is applied at the input and output layers only, which will be discussed in Section 4.2.3. As a result, this RevIN procedure can be considered to first make problems easier, and then restore them back to the original state, rather than directly solving the challenging problem where the distribution shift problem exists.

## 4 EXPERIMENTS

This section describes the experimental setup and provides extensive experimental results of RevIN.

### 4.1 EXPERIMENTAL SETUP

**Datasets.** We evaluate our methods mainly on four large-scale real-world time-series datasets. Additionally, we provide experimental results on three more datasets, including the air quality and Nasdaq datasets taken from the UCI repository and M4 competition dataset (Makridakis et al., 2020) in Appendix A.1. **(i) Electricity transformer temperature (ETT)**[1] data consists of seven features, including power load features and oil temperature. It is collected from two different regions in China for two years. Following the same protocol as Informer (Zhou et al., 2021), we split the data into three datasets: $ETTh_1$, $ETTh_2$, and $ETTm_1$. The $ETTh_1$ and $ETTh_2$ datasets are hourly data obtained from different regions. The $ETTm_1$ dataset has a value every 15 minutes. For each dataset, we split the first 12 months, the middle four months, and the last four months as training, validation, and test data, respectively. **(ii) Electricity Consuming Load (ECL)**[2] data contains the electricity consumption (kWh) collected from 321 clients. Following the prior work (Zhou et al., 2021), data from each client is used as a variable on an hourly basis in the multivariate forecasting setting. For the ECL dataset, we use 15, 3, and 4 months as training, validation, and test data, respectively.

**Experimental details.** We set the prediction lengths to be one day (1d), 2d, 7d, 14d, 30d, and 40d for the hourly-basis datasets, $ETTh_1$, $ETTh_2$, and ECL. For the $ETTm_1$ dataset, we chose six hours (6h), 12h, 3d, 7d, and 14d as the prediction window lengths. We evaluate the time-series forecasting performance on the mean squared error (MSE) and mean absolute error (MAE). Following the same evaluation procedure used in the previous study (Zhou et al., 2021), we compute the MSE and MAE on z-score normalized data to measure different variables on the same scale. More details on experimental settings, including training details and hyperparameters, are provided in Appendix A.11.

**Baselines compared.** RevIN is a model-agnostic method, generally applicable to any deep neural network. In this paper, we verify the effectiveness of RevIN by adopting it to three state-of-the-art time-series forecasting models: Informer (Zhou et al., 2021), N-BEATS (Oreshkin et al., 2020), and SCINet (Liu et al., 2021). These are non-autoregressive forecasting models. The reproduction details for the baselines are provided in Appendix A.12. Unless stated otherwise, we compare RevIN and the baselines under the same hyperparameter settings, including the input and prediction lengths.

### 4.2 RESULTS AND ANALYSES

This section provides the quantitative analysis and qualitative visualization results of RevIN in comparison with the state-of-the-art time-series forecasting baselines.

#### 4.2.1 EFFECTIVENESS OF REVERSIBLE INSTANCE NORMALIZATION ON VARIOUS TIME-SERIES FORECASTING MODELS

Table 1 compares the forecasting accuracy of the baselines and RevIN. The results show that RevIN consistently outperforms all three baselines, Informer, N-BEATS, and SCINet, by a large margin,

---

[1]https://github.com/zhouhaoyi/ETDataset
[2]https://archive.ics.uci.edu/ml/datasets/ElectricityLoadDiagrams20112014

Table 1: **Comparison of forecasting errors between the baselines and RevIN.** The analysis on the four datasets, ETTh$_1$, ETTh$_2$, ETTm$_1$, and ECL, is conducted by increasing the prediction length from 24 to 960/1344. We report the average errors for five runs. The complete results are provided in Appendix A.14, including standard deviation and the originally reported values for the baselines.

| Method | | Informer | | + RevIN | | N-BEATS | | + RevIN | | SCINet | | + RevIN | |
|---|---|---|---|---|---|---|---|---|---|---|---|---|---|
| Metric | | MSE | MAE | MSE | MAE | MSE | MAE | MSE | MAE | MSE | MAE | MSE | MAE |
| ETTh$_1$ | 24 | 0.550 | 0.536 | **0.504** | **0.472** | 0.478 | 0.505 | **0.330** | **0.373** | 0.338 | 0.373 | **0.308** | **0.347** |
| | 48 | 0.772 | 0.668 | **0.646** | **0.547** | 0.536 | 0.542 | **0.372** | **0.400** | 0.436 | 0.459 | **0.365** | **0.389** |
| | 168 | 1.138 | 0.853 | **0.655** | **0.561** | 1.005 | 0.782 | **0.466** | **0.452** | 0.459 | 0.461 | **0.406** | **0.416** |
| | 336 | 1.278 | 0.909 | **1.058** | **0.758** | 0.932 | 0.743 | **0.515** | **0.483** | 0.527 | 0.513 | **0.467** | **0.471** |
| | 720 | 1.357 | 0.945 | **0.926** | **0.717** | 1.389 | 0.926 | **0.576** | **0.534** | 0.596 | 0.571 | **0.507** | **0.505** |
| | 960 | 1.470 | 0.990 | **0.902** | **0.715** | 1.383 | 0.932 | **0.678** | **0.575** | 0.604 | 0.574 | **0.545** | **0.526** |
| ETTh$_2$ | 24 | 0.450 | 0.520 | **0.238** | **0.325** | 0.403 | 0.472 | **0.192** | **0.276** | 0.199 | 0.295 | **0.180** | **0.263** |
| | 48 | 2.171 | 1.200 | **0.361** | **0.404** | 1.330 | 0.918 | **0.254** | **0.320** | 0.350 | 0.422 | **0.231** | **0.302** |
| | 168 | 8.157 | 2.558 | **0.859** | **0.649** | 7.174 | 2.329 | **0.410** | **0.418** | 0.559 | 0.518 | **0.337** | **0.378** |
| | 336 | 4.746 | 1.844 | **0.890** | **0.673** | 4.859 | 1.863 | **0.449** | **0.447** | 0.664 | 0.583 | **0.357** | **0.403** |
| | 720 | 3.190 | 1.529 | **0.576** | **0.546** | 5.656 | 2.012 | **0.496** | **0.482** | 1.546 | 0.944 | **0.411** | **0.445** |
| | 960 | 2.972 | 1.441 | **0.600** | **0.570** | 6.408 | 2.077 | **0.471** | **0.481** | 1.862 | 1.066 | **0.438** | **0.462** |
| ETTm$_1$ | 24 | 0.330 | 0.382 | **0.309** | **0.352** | 0.443 | 0.437 | **0.403** | **0.392** | 0.130 | 0.231 | **0.106** | **0.196** |
| | 48 | 0.499 | 0.486 | **0.390** | **0.391** | 0.453 | 0.472 | **0.328** | **0.371** | 0.155 | 0.262 | **0.135** | **0.222** |
| | 96 | 0.605 | 0.554 | **0.405** | **0.411** | 0.603 | 0.581 | **0.379** | **0.406** | 0.195 | 0.291 | **0.162** | **0.247** |
| | 288 | 0.906 | 0.738 | **0.563** | **0.502** | 0.849 | 0.702 | **0.451** | **0.445** | 0.361 | 0.419 | **0.265** | **0.321** |
| | 672 | 0.943 | 0.760 | **0.663** | **0.550** | 0.860 | 0.726 | **0.555** | **0.511** | 1.020 | 0.756 | **0.357** | **0.380** |
| | 1344 | 1.095 | 0.823 | **0.824** | **0.632** | 14.613 | 1.948 | **0.631** | **0.556** | 1.841 | 1.044 | **0.412** | **0.422** |
| ECL | 24 | 0.250 | 0.358 | **0.148** | **0.257** | 0.279 | 0.372 | **0.176** | **0.285** | 0.138 | 0.246 | **0.112** | **0.207** |
| | 48 | 0.300 | 0.386 | **0.171** | **0.279** | 0.309 | 0.388 | **0.194** | **0.301** | 0.163 | 0.265 | **0.126** | **0.222** |
| | 168 | 0.345 | 0.423 | **0.261** | **0.354** | 0.333 | 0.410 | **0.218** | **0.320** | 0.177 | 0.281 | **0.153** | **0.249** |
| | 336 | 0.429 | 0.473 | **0.356** | **0.414** | 0.326 | 0.406 | **0.241** | **0.337** | 0.202 | 0.308 | **0.162** | **0.262** |
| | 720 | 0.851 | 0.719 | **0.834** | **0.700** | 0.420 | 0.467 | **0.303** | **0.383** | 0.234 | 0.333 | **0.183** | **0.281** |
| | 960 | 0.930 | 0.750 | **0.894** | **0.741** | 0.399 | 0.455 | **0.325** | **0.398** | 0.235 | 0.330 | **0.200** | **0.292** |

Table 2: **Comparison of long sequence forecasting performance.** We analyze the forecasting error of the baselines and RevIN by increasing the prediction length from 48 to 960 while the input length is fixed to 48. The experiment is conducted on ETTh$_1$. The average errors for five runs are reported, and the complete results, including standard deviation, are provided in Appendix A.14.

| Prediction length | | 48 | | 168 | | 336 | | 720 | | 960 |
|---|---|---|---|---|---|---|---|---|---|---|
| Metric | MSE | MAE | MSE | MAE | MSE | MAE | MSE | MAE | MSE | MAE |
| Informer | 0.687 | 0.628 | 0.982 | 0.795 | 1.212 | 0.893 | 1.157 | 0.863 | 1.203 | 0.888 |
| **+ RevIN** | **0.540** | **0.481** | **0.680** | **0.574** | **0.939** | **0.696** | **1.021** | **0.752** | **1.061** | **0.775** |
| N-BEATS | 0.512 | 0.523 | 0.804 | 0.690 | 1.001 | 0.773 | 1.022 | 0.765 | 0.901 | 0.728 |
| **+ RevIN** | **0.365** | **0.389** | **0.454** | **0.438** | **0.526** | **0.477** | **0.568** | **0.514** | **0.638** | **0.544** |
| SCINet | 0.376 | 0.396 | 0.600 | 0.556 | 0.841 | 0.695 | 0.875 | 0.721 | 0.900 | 0.737 |
| **+ RevIN** | **0.349** | **0.370** | **0.445** | **0.426** | **0.509** | **0.461** | **0.533** | **0.494** | **0.557** | **0.510** |

achieving state-of-the-art performance on the four datasets. Moreover, the effectiveness of RevIN is more evident for the long sequence prediction, where it remarkably reduces the errors of the baselines. RevIN shows a stable performance in contrast to the baselines, which show a high increase in error as prolonging the prediction length. For example, when the prediction length increases from 24 to 960 on the ETTh$_2$ dataset, the forecasting error of N-BEATS significantly increases from 0.403 to 6.408. In contrast, RevIN shows a much slight increase in error, i.e., from 0.192 to 0.471. A similar tendency appears with the other prediction lengths, datasets, and baseline models as well. These results demonstrate that RevIN makes the baseline model more robust to prediction length.

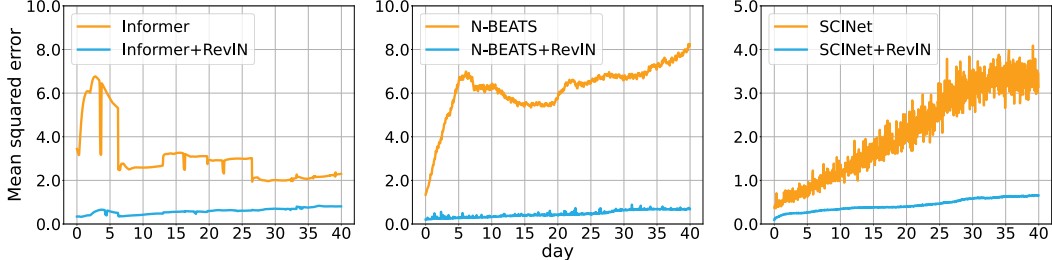

Figure 4: **Forecasting error for each time step.** We compare the error of predicting 1∼960 steps ahead between the baselines and RevIN on ETTh$_1$ when the prediction length is 960 (40 days).

Table 3: **Comparison with classical and state-of-the-art normalization methods.** The mean squared errors are compared on the four datasets, using N-BEATS as the baseline for all experiments. Here, every normalization method is applied to the input data. DAIN, deep adaptive input normalization (Passalis et al., 2019); RevBN, the reversible batch normalization, i.e., the modified version of RevIN. Complete results, including different prediction lengths, are provided in Appendix A.7.

| Dataset | ETTh$_1$ | | ETTh$_2$ | | ETTm$_1$ | | ECL | |
|---|---|---|---|---|---|---|---|---|
| Prediction length | 168 | 960 | 168 | 960 | 96 | 1344 | 168 | 960 |
| Min-max norm | 1.074 | 1.224 | 2.987 | 3.308 | 1.035 | 1.320 | 0.374 | 0.387 |
| z-score norm | 0.953 | 1.043 | 3.329 | 3.087 | 1.016 | 1.274 | 0.335 | 0.377 |
| Layer norm | 0.871 | 1.303 | 4.092 | 5.822 | 0.502 | 2.488 | 0.343 | 0.379 |
| DAIN | 0.996 | 1.032 | 1.982 | 2.802 | 0.672 | 1.348 | 0.347 | 0.381 |
| Batch norm | 0.851 | 1.691 | 6.206 | 7.755 | 0.505 | 1.147 | 0.320 | 0.422 |
| **RevBN** | 0.717 | 0.779 | 0.729 | 2.148 | 0.601 | 0.991 | 0.327 | 0.414 |
| Instance norm | 0.946 | 1.090 | 3.240 | 3.145 | 1.021 | 1.329 | 0.333 | 0.386 |
| **RevIN (ours)** | **0.515** | **0.697** | **0.419** | **0.465** | **0.388** | **0.602** | **0.220** | **0.329** |

We further quantitatively analyze the effect of RevIN on long sequence prediction in Table 2. When the model prediction length is increased from 48 to 960, RevIN reduces the prediction error compared to the baseline, showing robust performance against the prediction length. The difference in the forecasting error between SCINet and RevIN is relatively small when the prediction length is short (e.g., 48), but RevIN remarkably surpasses SCINet by a significant margin when the prediction length is long (e.g., 336, 720, and 960). These results substantiate that adopting RevIN can make a model robust to the prediction length.

Additionally, to study how RevIN can perform well in long sequence prediction, we visualize the forecasting error for each time step in Fig. 4. The error at the $t$-th time step is computed as MSE-$t = \frac{1}{N} \sum_{i=1}^{N} \frac{1}{K} \sum_{k=1}^{K} (\hat{y}_{kt}^{(i)} - y_{kt}^{(i)})^2$. Overall, RevIN shows superior performance compared to the baselines; the performance degradation is significantly slower, showing low error even when forecasting 960 steps ahead. Specifically, for N-BEATS and SCINet, the error extremely increases as predicting the distant future values. RevIN alleviates this significant increase in error, showing remarkable performance compared to the baselines. Informer also shows unstable performance for the different time steps. The error is substantial in the early steps and becomes relatively small in the distant steps. RevIN allows the models to have consistently minor errors in every time step, even where the baselines originally show high error (early steps for Informer, distant steps for N-BEATS and SCINet). The results demonstrate the effectiveness of RevIN when forecasting long sequences.

### 4.2.2 COMPARISON WITH EXISTING NORMALIZATION METHODS

We compare RevIN with classical and state-of-the-art normalization methods, including min-max normalization, z-score normalization, layer normalization (Ba et al., 2016), batch normalization (Ioffe & Szegedy, 2015), instance normalization (Ulyanov et al., 2016), and deep adaptive input

Figure 5: **Feature divergence between the training and test data in the intermediate layers of the model** The feature divergences are computed on the ETTh$_1$, ETTh$_2$, ETTm$_1$, and ECL datasets using the features obtained from the first (Layer-1) and the second (Layer-2) encoder layers in Informer.

normalization (DAIN) (Passalis et al., 2019) in Table 3. Here, we compute the statistics for min-max and z-score normalization methods for every input instance, not for the entire data. Additionally, we attempt to use batch normalization as the input normalization method in RevIN, named reversible batch normalization (RevBN). Layer normalization cannot be used in a similar manner since it is not reversible when the input and prediction lengths are different, as in our experimental settings.

As a result, RevIN shows outstanding performance compared to the other normalization methods, especially on ETTh$_2$ and ETTm$_1$ datasets. Also, RevBN improves forecasting performance of batch normalization. Specifically, the error considerably decreases in the long sequence prediction, such as 960 and 1344. This result supports that the denormalization step of RevIN is essential as a key component of the proposed method for improving long sequence forecasting. However, batch normalization applies identical normalization to all the input sequences, using the global statistics obtained from the entire training data; it can not reduce the discrepancy between the training and test data distributions. Consequently, RevIN, which transforms the data in the instance level, outperforms RevBN by a significant margin, demonstrating that successfully reducing the discrepancy between distributions of different input sequences can effectively improve performance. Moreover, RevIN not only shows the best performance but also has the advantage of being lightweight compared to the baselines. For example, when $K$ is the number of variables, DAIN requires at least $3K^2$ additional parameters, whereas RevIN only requires $2K$ additional parameters.

### 4.2.3 ANALYSIS OF DISTRIBUTION SHIFT IN THE INTERMEDIATE LAYERS

In Fig. 5, we analyze the feature divergence between the training and test data to verify that RevIN can reduce the distribution shift at the intermediate feature level as well. We conduct the experiment using Informer as the baseline; it comprises two encoder layers and one decoder layer. Thus, we analyze the features of the first (Layer-1) and the second (Layer-2) encoder layers. Following the prior work (Pan et al., 2018), we compute the average feature divergence using symmetric KL divergence (See Appendix A.10). The results show that RevIN significantly reduces the feature divergence between the training and test data in both layers, demonstrating that the proposed approach, when added only to the input and output layers, successfully alleviates the distribution shift problem in the intermediate layers. Moreover, this strengthens RevIN as a generally-applicable flexible layer. An arbitrary model can adopt RevIN by adding it to input and output layers without any architectural modifications. Note that our approach still can be added to any arbitrarily chosen layers, significantly improving the model performance, as shown in Appendix A.4.

## 5 CONCLUSION

This paper aims to address the distribution shift problem in time series, proposing a simple yet effective normalization-and-denormalization method, reversible instance normalization (RevIN). The proposed approach effectively alleviates the discrepancy between training and test data distributions, leading to significant performance improvements in time-series forecasting. As a generally-applicable layer to arbitrary deep neural networks, the proposed approach achieves state-of-the-art performance on seven real-world time-series datasets by a significant margin. The extensive quantitative and qualitative experiments with in-depth analysis demonstrate the effectiveness of RevIN for accurate time-series forecasting against the distribution shift problem.

## REPRODUCIBILITY STATEMENT

To ensure reproducibility, we provide the source code of our method publicly, including the pre-trained model weights. In the main manuscript, Section 4.1 describes how we conduct data preprocessing on the datasets used in the experiments. Appendix A.11 explains the experimental details, including random seed values for the experiments. Appendix A.12 provides a detailed explanation of hyperparameter configurations with the reproduction details of the baselines.

## ACKNOWLEDGMENTS

This work was supported by the Institute of Information & communications Technology Planning & Evaluation (IITP) grant funded by the Korea government (MSIT) (No. 2018-0-00219, Space-time complex artificial intelligence blue-green algae prediction technology based on direct-readable water quality complex sensor and hyperspectral image, and No.2019-0-00075, Artificial Intelligence Graduate School Program (KAIST)).

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

# A  APPENDIX

This section provides additional information, visualizations, and experimental results that support the main manuscript. Section A.1 shows the experimental results on additional real-world datasets along with a qualitative analysis on one of the datasets to verify the effectiveness of our method on obvious non-stationary time series. Section A.2 addresses the potential of RevIN on solving the cross-domain time-series forecasting task. Section A.3 and Section A.4 present the hyperparameter sensitivity analysis and the ablation study on the proposed method, respectively. Section A.5 evaluates our method using complementary metrics, which measure the similarity between two sequences. Section A.6 and Section A.7 compare the forecasting results using the proposed method and existing normalization methods. Section A.8, Section A.9, and Section A.10 provide the algorithm for RevIN, the theoretical justification of RevIN, and the calculation details of the feature divergence used in Section 4.2.3, respectively. Section A.11 and Section A.12 describe additional implementation details and the reproduction details of the baselines where RevIN is applied, respectively. Section A.13 illustrates additional quantitative results for RevIN and the baselines. Lastly, Section A.14 provides complete quantitative results, including the standard deviation values for five experiments, which can not be included in the main manuscript due to the lack of space.

## A.1  EXPERIMENTAL RESULTS ON ADDITIONAL REAL-WORLD TIME-SERIES DATASETS

Table 4: **Forecasting performance on the air quality, Nasdaq, and M4 competition datasets. The results on the M4 dataset (\*) are multiplied by ten for readability. The average value and the standard deviation value for five runs are reported.**

| Methods | | Informer | | + RevIN | | N-BEATS | | + RevIN | | SCINet | | + RevIN | |
|---|---|---|---|---|---|---|---|---|---|---|---|---|---|
| Metric | | MSE | MAE | MSE | MAE | MSE | MAE | MSE | MAE | MSE | MAE | MSE | MAE |
| Air quality | 24 | 0.802 ± 0.178 | 0.671 ± 0.084 | **0.585** ± **0.033** | **0.539** ± **0.023** | 0.698 ± 0.064 | 0.626 ± 0.029 | **0.527** ± **0.005** | **0.498** ± **0.003** | 0.512 ± 0.029 | 0.514 ± 0.019 | **0.490** ± **0.006** | **0.474** ± **0.004** |
| | 48 | 0.966 ± 0.054 | 0.761 ± 0.023 | **0.859** ± **0.086** | **0.668** ± **0.041** | 0.955 ± 0.106 | 0.740 ± 0.035 | **0.705** ± **0.019** | **0.600** ± **0.009** | 0.712 ± 0.091 | 0.627 ± 0.047 | **0.659** ± **0.013** | **0.566** ± **0.007** |
| | 168 | 1.328 ± 0.107 | 0.923 ± 0.040 | **1.036** ± **0.056** | **0.761** ± **0.020** | 1.079 ± 0.108 | 0.818 ± 0.046 | **0.789** ± **0.008** | **0.660** ± **0.005** | 0.957 ± 0.067 | 0.737 ± 0.031 | **0.794** ± **0.025** | **0.645** ± **0.014** |
| | 336 | 1.278 ± 0.074 | 0.901 ± 0.032 | **1.145** ± **0.032** | **0.801** ± **0.010** | 1.105 ± 0.052 | 0.835 ± 0.021 | **0.860** ± **0.017** | **0.685** ± **0.006** | 0.989 ± 0.111 | 0.760 ± 0.046 | **0.854** ± **0.029** | **0.676** ± **0.010** |
| | 720 | 2.028 ± 0.216 | 1.104 ± 0.061 | **1.161** ± **0.028** | **0.810** ± **0.009** | 1.538 ± 0.419 | 0.968 ± 0.110 | **0.842** ± **0.015** | **0.686** ± **0.008** | 1.228 ± 0.048 | 0.858 ± 0.021 | **0.839** ± **0.024** | **0.680** ± **0.013** |
| Nasdaq | 30 | 5.318 ± 0.052 | 1.093 ± 0.017 | **1.273** ± **0.078** | **0.630** ± **0.009** | 5.500 ± 0.647 | 1.254 ± 0.086 | **1.023** ± **0.034** | **0.577** ± **0.007** | 1.742 ± 0.111 | 0.739 ± 0.028 | **0.985** ± **0.018** | **0.564** ± **0.005** |
| | 60 | 5.525 ± 0.022 | 1.098 ± 0.016 | **1.573** ± **0.098** | **0.666** ± **0.011** | 5.226 ± 0.424 | 1.236 ± 0.032 | **1.207** ± **0.044** | **0.617** ± **0.009** | 2.304 ± 0.062 | 0.790 ± 0.010 | **1.161** ± **0.021** | **0.601** ± **0.003** |
| | 120 | 5.793 ± 0.140 | 1.090 ± 0.012 | **2.648** ± **0.186** | **0.762** ± **0.016** | 6.023 ± 0.382 | 1.197 ± 0.034 | **1.959** ± **0.062** | **0.714** ± **0.006** | 3.227 ± 0.236 | 0.853 ± 0.007 | **1.869** ± **0.037** | **0.697** ± **0.003** |
| M4* | average | 0.099 ± 0.002 | 0.258 ± 0.020 | **0.008** ± **0.005** | **0.074** ± **0.005** | 2.241 ± 0.037 | 2.065 ± 0.029 | **2.082** ± **0.014** | **1.974** ± **0.006** | 2.180 ± 1.943 | 1.943 ± 0.011 | **2.079** ± **0.011** | **1.892** ± **0.004** |

We evaluate the proposed method on the four large-scale real-world time-series datasets, the ETTh$_1$, ETTh$_2$, ETTm$_1$, and ECL datasets in the main manuscript. Additionally, this section provides experimental results on three more datasets, including two real-world datasets taken from the UCI repository, the air quality dataset and the Nasdaq dataset, and the M4 competition dataset (Makridakis et al., 2020). In total, our proposed method is evaluated on seven datasets in this paper.

**Air quality**[3] dataset contains hourly averaged responses collected from five metal oxide chemical sensors located in Italy. The data consist of 13 variables of length 9537. We set the prediction length as {24, 48, 168, 336, 720} and the corresponding input length as {48, 96, 168, 168, 360} so that their ratios become {2x, 2x, 1x, 0.5x, 0.5x}.

**Nasdaq**[4] dataset consists of 82 variables, including important indices of markets around the world, the price of major companies in the U.S. market, treasury bill rates, etc. It is measured daily, having a total of 1984 data samples for each variable. We prolong the prediction length as {30, 60, 120} and set the corresponding input length as 60 for all.

---

[3]https://archive.ics.uci.edu/ml/datasets/Air+Quality

[4]https://archive.ics.uci.edu/ml/datasets/CNNpred%3A+CNN-based+stock+market+prediction+using+a+diverse+set+of+variables

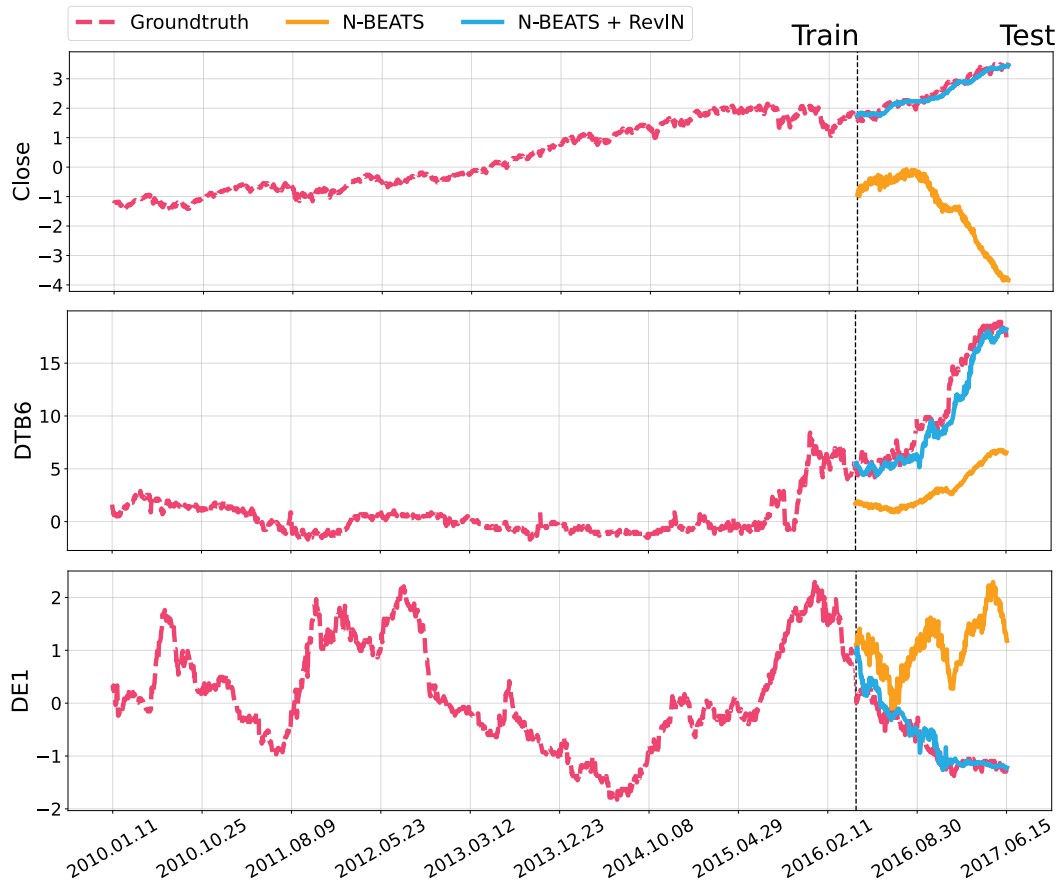

Figure 6: **Prediction results on the Nasdaq dataset.** The results on three variables in the data, Close, DTB6, and DE1, are shown. The prediction length is 60 days, and the seventh value is illustrated to show the results on the entire test set. We compare RevIN with N-BEATS.

**M4**[5] competition dataset consists of six different hourly, daily, weekly, monthly, quarterly, and yearly sets, containing 100,000 test data. We follow the original experimental protocol of the M4 competition (Makridakis et al., 2020). For evaluation, we measure the micro-averaged mean absolute error and mean squared error: we first compute the metric independently for each set, i.e., hourly, daily, weekly, monthly, quarterly, and yearly sets, and then calculate the weighted average of the metrics using the contributions of each set as the weights.

As shown in Table 4, RevIN significantly improves the forecasting performance of the baselines on all three datasets. Notably, RevIN shows outstanding performance on the Nasdaq dataset, reducing the prediction errors by more than half compared to the baselines.

Additionally, we conduct a qualitative analysis on the Nasdaq dataset to verify the effectiveness of our method on obvious non-stationary time series. As shown in Fig. 6, the Nasdaq index (the variable 'Close') has steadily increased since 2010. Accordingly, when data are divided into the training and test data based on a specific point in time (vertical dashed line in Fig. 6), the test data values tend to be higher than the training data values. In other words, the data severely suffers from the distribution shift problem, where the training and test data show a discrepancy in their distribution. As a result, even existing state-of-the-art models often cannot predict the future values appropriately, as shown in Fig. 6. The baseline fails to keep up with the trend in data, whose mean value continues to increase, and thus the prediction results are shifted. However, RevIN mitigates this distribution discrepancy and remarkably increases the prediction performance of the baseline.

---

[5]https://mofc.unic.ac.cy/m4/

Similarly, RevIN shows superior performance on the other rapidly increasing data ('DTB6' in Fig. 6) and the decreasing test data ('DE1' in Fig. 6), accurately predicting the changing mean of the data.

## A.2   CROSS-DOMAIN TIME-SERIES FORECASTING

Table 5: **Cross-domain time-series forecasting results.** We conduct a cross-domain evaluation on the ETT datasets, $ETTh_1$, $ETTh_2$, and $ETTm_1$. We train RevIN using SCINet as the baseline. We report the average errors and the standard deviation values for five runs.

| Train | | $ETTh_1$ | | | | $ETTh_2$ | | | | $ETTm_1$ | | | |
|---|---|---|---|---|---|---|---|---|---|---|---|---|---|
| Test | | $ETTh_2$ | | $ETTm_1$ | | $ETTh_1$ | | $ETTm_1$ | | $ETTh_1$ | | $ETTh_2$ | |
| Prediction length | | 336 | 960 | 336 | 960 | 336 | 960 | 336 | 960 | 288 | 1344 | 288 | 1344 |
| SCINet | MSE | 0.471 $\pm 0.034$ | 0.741 $\pm 0.056$ | 0.471 $\pm 0.043$ | 0.765 $\pm 0.029$ | 0.614 $\pm 0.020$ | 0.671 $\pm 0.061$ | 0.608 $\pm 0.049$ | 1.668 $\pm 0.110$ | 0.659 $\pm 0.032$ | 0.654 $\pm 0.026$ | 0.518 $\pm 0.009$ | 1.813 $\pm 0.166$ |
| | MAE | 0.450 $\pm 0.024$ | 0.640 $\pm 0.027$ | 0.427 $\pm 0.032$ | 0.636 $\pm 0.013$ | 0.507 $\pm 0.014$ | 0.607 $\pm 0.035$ | 0.487 $\pm 0.031$ | 1.004 $\pm 0.046$ | 0.562 $\pm 0.015$ | 0.608 $\pm 0.016$ | 0.509 $\pm 0.004$ | 1.027 $\pm 0.078$ |
| **+ RevIN** | MSE | **0.350** $\pm$ **0.010** | **0.419** $\pm$ **0.007** | **0.346** $\pm$ **0.008** | **0.452** $\pm$ **0.004** | **0.501** $\pm$ **0.006** | **0.586** $\pm$ **0.015** | **0.321** $\pm$ **0.009** | **0.441** $\pm$ **0.001** | **0.478** $\pm$ **0.003** | **0.542** $\pm$ **0.013** | **0.387** $\pm$ **0.005** | **0.506** $\pm$ **0.010** |
| | MAE | **0.383** $\pm$ **0.006** | **0.451** $\pm$ **0.004** | **0.331** $\pm$ **0.005** | **0.449** $\pm$ **0.001** | **0.449** $\pm$ **0.004** | **0.542** $\pm$ **0.007** | **0.331** $\pm$ **0.004** | **0.445** $\pm$ **0.001** | **0.477** $\pm$ **0.002** | **0.531** $\pm$ **0.006** | **0.417** $\pm$ **0.002** | **0.503** $\pm$ **0.006** |

As RevIN can alleviate the distribution discrepancy between the training and test data, we investigate the ability of RevIN in mitigating distribution discrepancy between different domains through cross-domain time-series forecasting task. In time series, a domain can be a location of a sensor where the data is collected. We use the ETT datasets for the experiment since they have the same feature categories. However, their distributions can exhibit significant discrepancy because the $ETTh_1$ and $ETTh_2$ datasets are collected from different locations, and the ETTh and $ETTm_1$ datasets have different measurement time intervals. Thus, we alternately use each ETT dataset as a source domain for training and a target domain for testing. The goal of cross-domain time-series forecasting is to alleviate the data distribution discrepancy between the source and target domains, e.g., between the $ETTh_1$ and $ETTh_2$ datasets.

In Table 5, despite the difference in the data distributions, RevIN shows remarkable performance in cross-domain time-series forecasting. In particular, RevIN outperforms SCINet by a large margin when the model needs to reduce the discrepancy between the $ETTh_2$ and $ETTm_1$ datasets. The results demonstrate that RevIN successfully solves the distribution shift problem by alleviating data distribution discrepancy between different domains, leading to better generalization performance.

## A.3   HYPERPARAMETER SENSITIVITY ANALYSIS

We analyze the hyperparameter sensitivity of the proposed method compared with the baseline models. Input sequence length can be a crucial hyperparameter to RevIN since the method computes the mean and the standard deviation across the entire input sequence and then uses the statistics at its

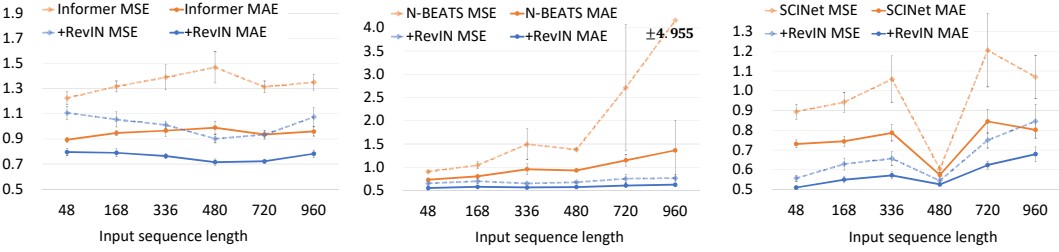

Figure 7: **Impacts of the input sequence length on RevIN compared with the baselines.** We prolong the input length from 48 (two days) to 960 (40 days) when the prediction length is set as 960 (40 days) on the $ETTh_1$ dataset. The average errors for five runs, with standard deviation values, are reported. In N-BEATS, the standard deviation value for the mean squared error is too large to visualize when the prediction length is 960; we write the value as "$\pm 4.955$" instead.

normalization and denormalization steps. Thus, the input sequence length would play a crucial role in the prediction accuracy and stability of the training process. Accordingly, we prolong the model input sequence length to see its impact on the forecasting performance of RevIN, as shown in Fig. 7. As a result, RevIN consistently outperforms the baseline for various input sequence lengths. More importantly, RevIN makes the baseline models more robust to the input length. In other words, RevIN shows stable performance for various input lengths in contrast to the baseline models, which shows the high variance in their performance according to the input length. Notably, in N-BEATS, the average error, as well as the standard deviation of the error, significantly increase as prolonging the input length. This is because the trend in data is expressed as a linear function in N-BEATS. As the input length becomes prolonged, there is a chance that the variance (or non-stationarity) in the time-series values will become higher, decreasing the accuracy of the linear trend to fit the data unless the data is monotonically increasing or decreasing. Also, the mispredicted trends linearly increase the error in future values. However, when N-BEATS adopts the RevIN layer, it removes the non-stationary statistics from the input, and thus, the model shows robust performance against the input length. Removing the variability, i.e., normalizing its mean and standard deviation, before feeding it to the model and returning it to the output makes the model learning stable.

## A.4 ABLATION STUDY

Table 6: **Ablation study results.** We ablate the affine transformation (affine.) from RevIN and evaluate forecasting performance on the six datasets. N-BEATS is used as the baseline for all experiments. We report the average and the standard deviation values for the five runs.

| Method | | + RevIN w/o affine. | | + RevIN w/ affine. (ours) | |
|---|---|---|---|---|---|
| Metric | | MSE | MAE | MSE | MAE |
| ETTh$_1$ | 48 | $0.370 \pm 0.006$ | $0.393 \pm 0.003$ | $\mathbf{0.363 \pm 0.005}$ | $\mathbf{0.389 \pm 0.003}$ |
| | 960 | $0.675 \pm 0.038$ | $0.576 \pm 0.014$ | $\mathbf{0.638 \pm 0.035}$ | $\mathbf{0.559 \pm 0.017}$ |
| ETTh$_2$ | 48 | $0.257 \pm 0.003$ | $0.322 \pm 0.002$ | $\mathbf{0.255 \pm 0.008}$ | $\mathbf{0.321 \pm 0.005}$ |
| | 960 | $0.483 \pm 0.012$ | $0.487 \pm 0.007$ | $\mathbf{0.471 \pm 0.015}$ | $\mathbf{0.481 \pm 0.008}$ |
| ETTm$_1$ | 96 | $0.384 \pm 0.013$ | $0.408 \pm 0.009$ | $\mathbf{0.378 \pm 0.011}$ | $\mathbf{0.406 \pm 0.007}$ |
| | 1344 | $0.664 \pm 0.085$ | $0.567 \pm 0.039$ | $\mathbf{0.631 \pm 0.061}$ | $\mathbf{0.556 \pm 0.020}$ |
| ECL | 48 | $0.197 \pm 0.002$ | $0.302 \pm 0.002$ | $\mathbf{0.195 \pm 0.002}$ | $\mathbf{0.301 \pm 0.001}$ |
| | 960 | $0.347 \pm 0.034$ | $0.415 \pm 0.026$ | $\mathbf{0.325 \pm 0.019}$ | $\mathbf{0.398 \pm 0.015}$ |
| Air quality | 48 | $0.707 \pm 0.009$ | $0.601 \pm 0.002$ | $\mathbf{0.705 \pm 0.019}$ | $\mathbf{0.600 \pm 0.009}$ |
| | 720 | $0.852 \pm 0.025$ | $0.689 \pm 0.013$ | $\mathbf{0.842 \pm 0.015}$ | $\mathbf{0.686 \pm 0.008}$ |
| Nasdaq | 30 | $0.983 \pm 0.017$ | $0.565 \pm 0.005$ | $\mathbf{0.981 \pm 0.017}$ | $\mathbf{0.564 \pm 0.005}$ |
| | 60 | $1.163 \pm 0.010$ | $0.610 \pm 0.002$ | $\mathbf{1.155 \pm 0.020}$ | $\mathbf{0.608 \pm 0.005}$ |

We ablate the affine transformation from RevIN to analyze its impact on forecasting performance. We conduct the analysis on the ETTh$_1$, ETTh$_2$, ETTm$_1$, ECL, Nasdaq, and air quality datasets using N-BEATS as the baseline. The results in Table 6 show that the affine transformation consistently contributes to performance improvement on a variety of datasets. As mentioned earlier, a model can add RevIN in an intermediate layer, even to several layers. While existing approaches are a preprocessing-and-postprocessing method applied outside of the main prediction model, RevIN is an end-to-end trainable layer that can be added to any layer in the model as batch normalization (Ioffe & Szegedy, 2015) and instance normalization (Ulyanov et al., 2016), which are recently proposed deep learning-based normalization layers. Thus, we verify that adopting RevIN in the intermediate layers instead of the input and output layers can improve the forecasting performance as well. We add RevIN to the first stack of N-BEATS and SCINet and evaluate their performance on the six datasets. The results in Table 7 demonstrate that even when added to the intermediate layers, RevIN improves the performance of the baselines, as a learnable normalization layer. We mainly focus on adding RevIN to the input and output of a model since it shows robust performance on average. Nevertheless, the model adopting RevIN in the intermediate layers consistently outperforms the baseline without RevIN, frequently achieving the best performance among all. This performance is even better than the dynamic normalization methods, LSTNet* and ES-RNN*, when they are

adopted to N-BEATS as well (See Table 9 in Appendix A.6). For example, when the prediction length is 960, the mean squared errors of LSTNet*, ES-RNN*, RevIN (inter.), and RevIN (i/o) are 5.627, 1.338, 0.523, and 0.471, on average. In conclusion, RevIN is a flexible, end-to-end trainable

Table 7: **Effectiveness of RevIN when added to the intermediate layers in the model.** We add RevIN to the first stack of N-BEATS and SCINet and evaluate their performance on the six datasets. We report the average value and standard deviation of five experiments. RevIN (inter.) indicates the model where RevIN is added to the intermediate layers of the baseline network. RevIN (i/o) indicates the model where RevIN is added to the input and output layer of the baseline network.

| Method | | N-BEATS | | + RevIN (inter.) | | + RevIN (i/o) | | SCINet | | + RevIN (inter.) | | + RevIN (i/o) | |
|---|---|---|---|---|---|---|---|---|---|---|---|---|---|
| Metric | | MSE | MAE | MSE | MAE | MSE | MAE | MSE | MAE | MSE | MAE | MSE | MAE |
| ETTh1 | 24 | 0.478 ± 0.022 | 0.505 ± 0.012 | 0.347 ± 0.006 | 0.389 ± 0.004 | **0.330** ± **0.006** | **0.373** ± **0.004** | 0.338 ± 0.012 | 0.373 ± 0.009 | **0.306** ± **0.004** | 0.347 ± 0.004 | 0.308 ± 0.003 | **0.347** ± **0.002** |
| | 48 | 0.536 ± 0.060 | 0.542 ± 0.041 | 0.375 ± 0.008 | 0.407 ± 0.005 | **0.372** ± **0.001** | **0.400** ± **0.002** | 0.436 ± 0.025 | 0.459 ± 0.021 | **0.363** ± **0.004** | 0.394 ± 0.004 | 0.365 ± 0.005 | **0.389** ± **0.003** |
| | 168 | 1.005 ± 0.146 | 0.782 ± 0.064 | 0.495 ± 0.086 | 0.481 ± 0.062 | **0.466** ± **0.030** | **0.452** ± **0.014** | 0.459 ± 0.015 | 0.461 ± 0.013 | 0.415 ± 0.001 | 0.424 ± 0.002 | **0.406** ± **0.003** | **0.416** ± **0.003** |
| | 336 | 0.932 ± 0.079 | 0.743 ± 0.042 | 0.538 ± 0.043 | 0.508 ± 0.027 | **0.515** ± **0.013** | **0.483** ± **0.008** | 0.527 ± 0.010 | 0.513 ± 0.006 | 0.552 ± 0.002 | 0.516 ± 0.001 | **0.467** ± **0.005** | **0.471** ± **0.003** |
| | 720 | 1.389 ± 0.230 | 0.926 ± 0.066 | 0.608 ± 0.016 | 0.572 ± 0.009 | **0.576** ± **0.035** | **0.534** ± **0.018** | 0.596 ± 0.015 | 0.571 ± 0.013 | 0.560 ± 0.007 | 0.550 ± 0.005 | **0.507** ± **0.006** | **0.505** ± **0.004** |
| | 960 | 1.383 ± 0.380 | 0.932 ± 0.120 | **0.664** ± **0.033** | 0.604 ± 0.015 | 0.678 ± 0.019 | **0.575** ± **0.009** | 0.604 ± 0.017 | 0.574 ± 0.014 | 0.619 ± 0.005 | 0.582 ± 0.002 | **0.545** ± **0.010** | **0.526** ± **0.005** |
| ETTh2 | 24 | 0.403 ± 0.185 | 0.472 ± 0.101 | 0.199 ± 0.002 | 0.291 ± 0.002 | **0.192** ± **0.003** | **0.276** ± **0.002** | 0.199 ± 0.026 | 0.295 ± 0.027 | 0.186 ± 0.003 | 0.272 ± 0.001 | **0.180** ± **0.004** | **0.263** ± **0.002** |
| | 48 | 1.330 ± 0.240 | 0.918 ± 0.073 | 0.263 ± 0.007 | 0.335 ± 0.005 | **0.254** ± **0.011** | **0.320** ± **0.008** | 0.350 ± 0.025 | 0.422 ± 0.027 | 0.313 ± 0.045 | 0.373 ± 0.032 | **0.231** ± **0.006** | **0.302** ± **0.006** |
| | 168 | 7.174 ± 0.449 | 2.329 ± 0.049 | 0.425 ± 0.015 | 0.434 ± 0.010 | **0.410** ± **0.010** | **0.418** ± **0.005** | 0.559 ± 0.044 | 0.518 ± 0.025 | 0.338 ± 0.003 | 0.380 ± 0.001 | **0.337** ± **0.007** | **0.378** ± **0.003** |
| | 336 | 4.859 ± 0.268 | 1.863 ± 0.043 | **0.446** ± **0.007** | 0.456 ± 0.005 | 0.449 ± 0.011 | **0.447** ± **0.006** | 0.664 ± 0.073 | 0.583 ± 0.030 | 0.422 ± 0.001 | 0.443 ± 0.001 | **0.357** ± **0.003** | **0.403** ± **0.002** |
| | 720 | 5.656 ± 1.053 | 2.012 ± 0.186 | 0.505 ± 0.022 | 0.501 ± 0.013 | **0.496** ± **0.008** | **0.482** ± **0.002** | 1.546 ± 0.378 | 0.944 ± 0.141 | 0.634 ± 0.010 | 0.564 ± 0.005 | **0.411** ± **0.003** | **0.445** ± **0.002** |
| | 960 | 6.408 ± 2.039 | 2.077 ± 0.242 | 0.523 ± 0.040 | 0.522 ± 0.025 | **0.471** ± **0.015** | **0.481** ± **0.008** | 1.862 ± 0.153 | 1.066 ± 0.055 | 0.734 ± 0.014 | 0.603 ± 0.005 | **0.438** ± **0.007** | **0.462** ± **0.004** |
| ETTm1 | 24 | 0.443 ± 0.043 | 0.437 ± 0.035 | **0.387** ± **0.018** | **0.391** ± **0.012** | 0.403 ± 0.006 | 0.392 ± 0.005 | 0.130 ± 0.003 | 0.231 ± 0.003 | 0.108 ± 0.002 | 0.203 ± 0.004 | **0.106** ± **0.002** | **0.196** ± **0.001** |
| | 48 | 0.453 ± 0.034 | 0.472 ± 0.018 | 0.341 ± 0.008 | 0.388 ± 0.007 | **0.328** ± **0.010** | **0.371** ± **0.007** | 0.155 ± 0.004 | 0.262 ± 0.004 | 0.142 ± 0.011 | 0.241 ± 0.013 | **0.135** ± **0.003** | **0.222** ± **0.002** |
| | 96 | 0.603 ± 0.051 | 0.581 ± 0.027 | 0.401 ± 0.007 | 0.428 ± 0.004 | **0.379** ± **0.011** | **0.406** ± **0.007** | 0.195 ± 0.012 | 0.291 ± 0.013 | 0.192 ± 0.016 | 0.285 ± 0.017 | **0.162** ± **0.001** | **0.247** ± **0.001** |
| | 288 | 0.849 ± 0.095 | 0.702 ± 0.051 | 0.502 ± 0.032 | 0.483 ± 0.018 | **0.451** ± **0.016** | **0.445** ± **0.008** | 0.361 ± 0.008 | 0.419 ± 0.004 | **0.264** ± **0.002** | 0.323 ± 0.001 | 0.265 ± 0.003 | **0.321** ± **0.002** |
| | 672 | 0.860 ± 0.057 | 0.726 ± 0.026 | **0.553** ± **0.020** | 0.512 ± 0.009 | 0.555 ± 0.011 | **0.511** ± **0.008** | 1.020 ± 0.081 | 0.756 ± 0.025 | 0.663 ± 0.081 | 0.583 ± 0.033 | **0.357** ± **0.004** | **0.380** ± **0.002** |
| | 1344 | 14.613 ± 26.108 | 1.948 ± 1.655 | 0.722 ± 0.064 | 0.594 ± 0.027 | **0.631** ± **0.061** | **0.556** ± **0.020** | 1.841 ± 0.242 | 1.044 ± 0.100 | 0.989 ± 0.211 | 0.717 ± 0.081 | **0.412** ± **0.008** | **0.422** ± **0.003** |
| ECL | 24 | 0.279 ± 0.007 | 0.372 ± 0.003 | 0.182 ± 0.001 | 0.300 ± 0.001 | **0.176** ± **0.002** | **0.285** ± **0.001** | 0.138 ± 0.004 | 0.246 ± 0.005 | **0.111** ± **0.000** | **0.207** ± **0.001** | 0.112 ± 0.001 | **0.207** ± **0.001** |
| | 48 | 0.309 ± 0.007 | 0.388 ± 0.004 | 0.207 ± 0.003 | 0.318 ± 0.002 | **0.194** ± **0.001** | **0.301** ± **0.001** | 0.163 ± 0.007 | 0.265 ± 0.007 | **0.124** ± **0.001** | **0.221** ± **0.001** | 0.126 ± 0.001 | 0.222 ± 0.001 |
| | 168 | 0.333 ± 0.016 | 0.410 ± 0.012 | 0.237 ± 0.007 | 0.340 ± 0.004 | **0.218** ± **0.002** | **0.320** ± **0.001** | 0.177 ± 0.003 | 0.281 ± 0.005 | 0.154 ± 0.002 | **0.248** ± **0.001** | 0.153 ± 0.003 | 0.249 ± 0.002 |
| | 336 | 0.326 ± 0.004 | 0.406 ± 0.001 | 0.245 ± 0.011 | 0.348 ± 0.007 | **0.241** ± **0.005** | **0.337** ± **0.002** | 0.202 ± 0.004 | 0.308 ± 0.004 | **0.161** ± **0.002** | **0.261** ± **0.002** | 0.162 ± 0.001 | 0.262 ± 0.001 |
| | 720 | 0.420 ± 0.094 | 0.467 ± 0.058 | 0.308 ± 0.019 | 0.393 ± 0.016 | **0.303** ± **0.012** | **0.383** ± **0.011** | 0.234 ± 0.006 | 0.333 ± 0.004 | 0.184 ± 0.003 | 0.283 ± 0.003 | **0.183** ± **0.003** | **0.281** ± **0.002** |
| | 960 | 0.399 ± 0.022 | 0.455 ± 0.017 | 0.335 ± 0.018 | 0.413 ± 0.015 | 0.325 ± 0.019 | 0.398 ± 0.015 | 0.235 ± 0.011 | 0.330 ± 0.008 | **0.196** ± **0.005** | 0.295 ± 0.005 | 0.200 ± 0.003 | **0.292** ± **0.002** |
| Air quality | 24 | 0.698 ± 0.064 | 0.626 ± 0.029 | 0.558 ± 0.011 | 0.537 ± 0.008 | **0.527** ± **0.005** | **0.498** ± **0.003** | 0.512 ± 0.029 | 0.514 ± 0.019 | **0.488** ± **0.006** | 0.486 ± 0.009 | 0.490 ± 0.006 | **0.474** ± **0.004** |
| | 48 | 0.955 ± 0.106 | 0.740 ± 0.035 | 0.722 ± 0.013 | 0.629 ± 0.005 | **0.705** ± **0.019** | **0.600** ± **0.009** | 0.712 ± 0.091 | 0.627 ± 0.047 | **0.651** ± **0.032** | 0.578 ± 0.023 | 0.659 ± 0.013 | **0.566** ± **0.007** |
| | 168 | 1.079 ± 0.108 | 0.818 ± 0.046 | 0.819 ± 0.007 | 0.691 ± 0.004 | **0.789** ± **0.008** | **0.660** ± **0.005** | 0.957 ± 0.067 | 0.737 ± 0.031 | **0.787** ± **0.020** | 0.648 ± 0.012 | 0.794 ± 0.025 | **0.645** ± **0.014** |
| | 336 | 1.105 ± 0.052 | 0.835 ± 0.021 | 0.902 ± 0.018 | 0.721 ± 0.010 | **0.860** ± **0.017** | **0.685** ± **0.006** | 0.989 ± 0.111 | 0.760 ± 0.046 | 0.870 ± 0.022 | 0.695 ± 0.011 | **0.854** ± **0.029** | **0.676** ± **0.010** |
| | 720 | 1.538 ± 0.419 | 0.968 ± 0.110 | 0.945 ± 0.030 | 0.757 ± 0.012 | **0.842** ± **0.015** | **0.686** ± **0.008** | 1.228 ± 0.048 | 0.858 ± 0.021 | 0.939 ± 0.064 | 0.730 ± 0.025 | **0.839** ± **0.024** | **0.680** ± **0.013** |
| Nasdaq | 30 | 5.500 ± 0.647 | 1.254 ± 0.086 | **0.940** ± **0.055** | 0.581 ± 0.037 | 1.023 ± 0.034 | 0.577 ± 0.007 | 1.742 ± 0.111 | 0.739 ± 0.028 | 1.111 ± 0.095 | 0.599 ± 0.020 | **0.985** ± **0.018** | **0.564** ± **0.005** |
| | 60 | 5.226 ± 0.424 | 1.236 ± 0.032 | **0.989** ± **0.025** | **0.578** ± **0.008** | 1.207 ± 0.044 | 0.617 ± 0.010 | 2.304 ± 0.062 | 0.790 ± 0.010 | 1.280 ± 0.023 | 0.630 ± 0.004 | **1.161** ± **0.021** | 0.601 ± 0.003 |
| | 120 | 6.023 ± 0.382 | 1.197 ± 0.034 | **1.166** ± **0.014** | **0.615** ± **0.003** | 1.959 ± 0.062 | 0.714 ± 0.006 | 3.227 ± 0.236 | 0.853 ± 0.007 | 2.585 ± 0.374 | 0.776 ± 0.031 | **1.869** ± **0.037** | **0.697** ± **0.003** |

layer that can significantly increase the performance of a model in time series forecasting, applied to any arbitrarily chosen layers.

## A.5 Performance Evaluation on Similarity Metrics for Time Series

We evaluate the forecasting performance of RevIN mainly on the mean squared error and the mean absolute error. Additionally, we use complementary metrics that measure the similarity between two sequences, dynamic time warping (DTW) and temporal distortion index (TDI) (Le Guen & Thome, 2019; Frías-Paredes et al., 2017). Table 8 shows that our approach significantly improves the baseline models across all datasets in terms of the DTW and TDI. Notably, RevIN exhibits outstanding performance by a large margin compared to the baselines for long prediction length. For example, when RevIN is added, the average DTW decreases from 38.348 to 15.240 for Informer, from 53.148 to 12.766 for N-BEATS, and from 20.498 to 11.080 for SCINet when the prediction length is 960 on ETTh$_2$. There are a few cases where the proposed method predicts a less similar sequence than the baseline. However, the margin is minimal in terms of either DTW or TDI compared to the significant margin found when our method outperforms the baseline. These results demonstrate that adopting RevIN can generate a sequence more similar to the groundtruth than the baseline, especially showing better prediction accuracy on the longer sequences.

Table 8: **Comparison results on similarity metrics for time series.** We assess the model forecasting results in terms of the shape and temporal errors using the DTW and the TDI, respectively (the lower, the better). The experiments are conducted for the four datasets, using the three baselines. We report the average value and standard deviation of five experiments.

| Method | | Informer | | + RevIN | | N-BEATS | | + RevIN | | SCINet | | + RevIN | |
|---|---|---|---|---|---|---|---|---|---|---|---|---|---|
| Metric | | TDI | DTW | TDI | DTW | TDI | DTW | TDI | DTW | TDI | DTW | TDI | DTW |
| ETTh$_1$ | 24 | 1.602 ± 0.093 | 2.515 ± 0.121 | **1.207** ± 0.037 | **2.206** ± 0.097 | 1.309 ± 0.147 | 2.361 ± 0.048 | **1.031** ± 0.021 | **1.830** ± 0.012 | 1.136 ± 0.113 | 1.784 ± 0.037 | **0.969** ± 0.017 | **1.672** ± 0.007 |
| | 48 | 3.932 ± 0.796 | 4.178 ± 0.333 | **2.762** ± 0.190 | **3.401** ± 0.063 | 2.430 ± 0.386 | 3.564 ± 0.230 | **1.592** ± 0.055 | **2.745** ± 0.012 | 2.418 ± 0.297 | 2.873 ± 0.117 | **1.456** ± 0.045 | **2.508** ± 0.012 |
| | 168 | 31.569 ± 4.652 | 10.384 ± 0.420 | **9.459** ± 1.173 | **6.532** ± 0.227 | 19.369 ± 3.679 | 8.937 ± 0.490 | **6.190** ± 1.033 | **5.756** ± 0.161 | 4.913 ± 1.085 | 5.344 ± 0.170 | **3.791** ± 0.075 | **5.017** ± 0.008 |
| | 336 | 72.959 ± 6.769 | 14.850 ± 0.514 | **45.215** ± 17.098 | **11.605** ± 1.321 | 47.961 ± 8.500 | 11.782 ± 0.531 | **14.302** ± 2.127 | **8.532** ± 0.162 | 6.120 ± 0.329 | 7.723 ± 0.087 | 6.973 ± 0.292 | **7.319** ± 0.025 |
| | 720 | 167.254 ± 13.008 | 23.295 ± 0.244 | **98.648** ± 18.571 | **18.483** ± 0.346 | 143.832 ± 28.984 | 20.494 ± 0.952 | **26.265** ± 8.439 | **12.383** ± 0.458 | 20.351 ± 3.588 | 11.550 ± 0.246 | **11.337** ± 0.296 | **10.511** ± 0.056 |
| | 960 | 182.008 ± 16.076 | 28.848 ± 1.217 | **128.152** ± 9.174 | **22.018** ± 0.204 | 148.671 ± 48.669 | 23.810 ± 2.418 | **40.774** ± 13.985 | **15.107** ± 0.744 | 24.067 ± 3.383 | 13.551 ± 0.270 | **14.148** ± 0.461 | **12.508** ± 0.063 |
| ETTh$_2$ | 24 | 2.016 ± 0.152 | 2.481 ± 0.353 | **1.528** ± 0.142 | **1.601** ± 0.029 | 1.476 ± 0.230 | 2.307 ± 0.488 | **1.128** ± 0.084 | **1.409** ± 0.007 | 1.210 ± 0.128 | 1.350 ± 0.095 | **1.088** ± 0.022 | **1.250** ± 0.015 |
| | 48 | **4.462** ± 0.567 | 8.194 ± 0.397 | 4.802 ± 0.342 | **2.806** ± 0.081 | 4.244 ± 0.470 | 6.131 ± 0.542 | **2.529** ± 0.165 | **2.207** ± 0.010 | 3.368 ± 0.380 | 2.362 ± 0.183 | **2.374** ± 0.058 | **1.906** ± 0.019 |
| | 168 | **24.016** ± 3.355 | 32.496 ± 1.563 | 24.516 ± 1.705 | **6.950** ± 0.313 | 42.270 ± 1.708 | 28.487 ± 0.642 | **14.759** ± 1.149 | **5.242** ± 0.104 | 12.307 ± 1.637 | 4.899 ± 0.229 | **10.205** ± 0.533 | **4.225** ± 0.083 |
| | 336 | 74.168 ± 9.015 | 29.549 ± 2.453 | **55.817** ± 3.180 | **9.572** ± 0.270 | 95.645 ± 6.432 | 30.837 ± 1.115 | **42.654** ± 3.450 | **8.063** ± 0.155 | 28.823 ± 2.681 | 7.447 ± 0.350 | **15.831** ± 0.141 | **5.975** ± 0.022 |
| | 720 | 129.440 ± 22.426 | 35.908 ± 2.230 | **111.842** ± 18.634 | **13.109** ± 0.391 | 185.321 ± 17.061 | 47.461 ± 6.122 | **92.521** ± 33.126 | **11.953** ± 0.977 | 142.118 ± 36.524 | 16.428 ± 2.748 | **26.577** ± 0.987 | **8.989** ± 0.035 |
| | 960 | 177.336 ± 18.126 | 38.348 ± 1.665 | **170.218** ± 14.187 | **15.240** ± 0.246 | 281.720 ± 75.629 | 53.148 ± 9.408 | **100.142** ± 29.742 | **12.766** ± 0.734 | 218.477 ± 27.365 | 20.498 ± 1.979 | **43.028** ± 2.007 | **11.080** ± 0.108 |
| ETTm$_1$ | 24 | **2.343** ± 0.130 | 1.730 ± 0.098 | 2.478 ± 0.094 | **1.590** ± 0.040 | **3.109** ± 0.135 | 2.062 ± 0.178 | 3.368 ± 0.067 | **1.882** ± 0.026 | 2.618 ± 0.096 | 1.194 ± 0.023 | **2.134** ± 0.034 | **0.995** ± 0.007 |
| | 48 | 4.818 ± 0.122 | 3.133 ± 0.097 | **4.116** ± 0.093 | **2.457** ± 0.038 | 5.223 ± 0.340 | 3.036 ± 0.083 | **4.400** ± 0.084 | **2.426** ± 0.061 | 4.176 ± 0.355 | 1.661 ± 0.077 | **3.317** ± 0.045 | **1.470** ± 0.015 |
| | 96 | 8.550 ± 0.671 | 4.813 ± 0.228 | **5.837** ± 0.098 | **3.586** ± 0.085 | 9.686 ± 1.050 | 5.220 ± 0.162 | **6.803** ± 0.271 | **3.803** ± 0.090 | 5.479 ± 0.349 | 2.409 ± 0.093 | **4.782** ± 0.045 | **2.180** ± 0.009 |
| | 288 | 32.918 ± 2.223 | 11.054 ± 0.341 | **17.231** ± 0.759 | **7.403** ± 0.122 | 42.809 ± 3.710 | 10.706 ± 0.513 | **15.942** ± 0.531 | **7.285** ± 0.064 | 22.650 ± 2.713 | 5.365 ± 0.072 | **15.457** ± 0.273 | **4.436** ± 0.033 |
| | 672 | 80.546 ± 17.739 | 16.365 ± 0.749 | **38.265** ± 6.209 | **12.030** ± 0.455 | 111.531 ± 13.075 | 16.601 ± 0.649 | **45.064** ± 5.369 | **12.482** ± 0.311 | 149.483 ± 10.180 | 13.980 ± 0.350 | **44.721** ± 1.867 | **8.040** ± 0.126 |
| | 1344 | 161.539 ± 16.808 | 23.822 ± 1.203 | **77.844** ± 7.652 | **17.293** ± 0.606 | 444.199 ± 59.765 | 69.125 ± 63.858 | **161.725** ± 99.382 | **19.490** ± 3.115 | 397.100 ± 39.675 | 27.319 ± 2.824 | **98.890** ± 4.495 | **12.779** ± 0.156 |
| ECL | 24 | 0.517 ± 0.007 | 1.610 ± 0.015 | **0.339** ± 0.005 | **1.234** ± 0.006 | 0.506 ± 0.004 | 1.702 ± 0.017 | **0.384** ± 0.002 | **1.368** ± 0.010 | 0.368 ± 0.017 | 1.171 ± 0.016 | **0.281** ± 0.001 | **1.058** ± 0.053 |
| | 48 | 0.677 ± 0.029 | 2.375 ± 0.038 | **0.383** ± 0.006 | **1.806** ± 0.012 | 0.676 ± 0.025 | 2.444 ± 0.028 | **0.446** ± 0.008 | **1.972** ± 0.011 | 0.466 ± 0.034 | 1.752 ± 0.038 | **0.318** ± 0.006 | **1.527** ± 0.004 |
| | 168 | 1.338 ± 0.017 | 4.364 ± 0.071 | **0.785** ± 0.024 | **3.709** ± 0.054 | 1.738 ± 0.166 | 4.576 ± 0.087 | **0.839** ± 0.013 | **3.828** ± 0.016 | 0.955 ± 0.077 | 3.379 ± 0.033 | **0.730** ± 0.029 | **3.122** ± 0.026 |
| | 336 | 2.276 ± 0.046 | 6.358 ± 0.148 | **1.495** ± 0.044 | **5.640** ± 0.126 | 3.555 ± 0.562 | 6.448 ± 0.049 | **1.661** ± 0.083 | **5.702** ± 0.038 | 2.336 ± 0.209 | 5.103 ± 0.033 | **1.194** ± 0.034 | **4.591** ± 0.012 |
| | 720 | 23.481 ± 15.963 | **16.561** ± 5.353 | 22.117 ± 12.615 | 17.227 ± 3.790 | 12.379 ± 4.196 | 10.841 ± 1.204 | **5.583** ± 1.060 | **9.239** ± 0.230 | 4.576 ± 0.806 | 7.913 ± 0.102 | **2.830** ± 0.092 | **7.162** ± 0.056 |
| | 960 | 32.548 ± 18.405 | **21.065** ± 3.649 | 28.328 ± 8.781 | 21.717 ± 3.002 | 15.866 ± 3.309 | 12.128 ± 0.377 | **9.198** ± 1.400 | **11.140** ± 0.347 | 6.972 ± 0.678 | 9.255 ± 0.108 | **4.502** ± 0.233 | **8.560** ± 0.064 |

Table 9: **Forecasting performance of RevIN in comparison with existing dynamic normalization methods.** LSTNet[*] indicates the model where the autoregressive linear bypass module of LSTNet is added to the baseline network. ES-RNN[*] indicates the model where the exponential smoothing of ES-RNN is added to the baseline network. The experiments are conducted on the ETTh$_1$, ETTh$_2$, ETTm$_1$, ECL, and the M4 datasets using N-BEATS as the baseline. The missing performances in the table are where the model fails to converge.

| Methods | | N-BEATS | | + LSTNet[*] | | + ESRNN[*] | | **+ RevIN** | |
|---|---|---|---|---|---|---|---|---|---|
| Metric | | MSE | MAE | MSE | MAE | MSE | MAE | MSE | MAE |
| ETTh$_1$ | 24 | 0.478 ± 0.022 | 0.505 ± 0.012 | 0.462 ± 0.047 | 0.497 ± 0.035 | 0.547 ± 0.031 | 0.515 ± 0.021 | **0.330 ± 0.006** | **0.373 ± 0.004** |
| | 48 | 0.536 ± 0.060 | 0.542 ± 0.041 | 0.587 ± 0.063 | 0.576 ± 0.043 | 0.662 ± 0.027 | 0.567 ± 0.012 | **0.372 ± 0.001** | **0.400 ± 0.002** |
| | 168 | 1.005 ± 0.146 | 0.782 ± 0.064 | 1.031 ± 0.099 | 0.795 ± 0.054 | 0.698 ± 0.044 | 0.600 ± 0.018 | **0.466 ± 0.030** | **0.452 ± 0.014** |
| | 336 | 0.932 ± 0.079 | 0.743 ± 0.042 | 0.964 ± 0.047 | 0.760 ± 0.023 | 0.768 ± 0.041 | 0.640 ± 0.022 | **0.515 ± 0.013** | **0.483 ± 0.008** |
| | 720 | 1.389 ± 0.230 | 0.926 ± 0.066 | 1.549 ± 0.061 | 0.994 ± 0.022 | 0.966 ± 0.084 | 0.742 ± 0.022 | **0.576 ± 0.035** | **0.534 ± 0.018** |
| | 960 | 1.383 ± 0.380 | 0.932 ± 0.120 | 1.293 ± 0.059 | 0.897 ± 0.026 | - | - | **0.678 ± 0.019** | **0.575 ± 0.009** |
| ETTh$_2$ | 24 | 0.403 ± 0.185 | 0.472 ± 0.101 | 0.394 ± 0.099 | 0.485 ± 0.068 | 0.614 ± 0.010 | 0.522 ± 0.004 | **0.192 ± 0.003** | **0.276 ± 0.002** |
| | 48 | 1.330 ± 0.240 | 0.918 ± 0.073 | 1.261 ± 0.214 | 0.907 ± 0.075 | 0.654 ± 0.009 | 0.543 ± 0.007 | **0.254 ± 0.011** | **0.320 ± 0.008** |
| | 168 | 7.174 ± 0.449 | 2.329 ± 0.049 | 7.053 ± 0.428 | 2.290 ± 0.095 | 0.962 ± 0.129 | 0.696 ± 0.054 | **0.410 ± 0.010** | **0.418 ± 0.005** |
| | 336 | 4.859 ± 0.268 | 1.863 ± 0.043 | 5.070 ± 0.336 | 1.914 ± 0.083 | 1.204 ± 0.158 | 0.789 ± 0.050 | **0.449 ± 0.011** | **0.447 ± 0.006** |
| | 720 | 5.656 ± 1.053 | 2.012 ± 0.186 | 6.311 ± 2.057 | 2.049 ± 0.225 | 1.284 ± 0.145 | 0.810 ± 0.033 | **0.496 ± 0.008** | **0.482 ± 0.002** |
| | 960 | 6.408 ± 2.039 | 2.077 ± 0.242 | 5.627 ± 1.670 | 1.965 ± 0.314 | 1.338 ± 0.535 | 0.809 ± 0.127 | **0.471 ± 0.015** | **0.481 ± 0.008** |
| ETTm$_1$ | 24 | 0.443 ± 0.043 | 0.437 ± 0.035 | 0.412 ± 0.026 | 0.426 ± 0.024 | 0.564 ± 0.015 | 0.477 ± 0.009 | **0.403 ± 0.006** | **0.392 ± 0.005** |
| | 48 | 0.453 ± 0.034 | 0.472 ± 0.018 | 0.420 ± 0.028 | 0.455 ± 0.018 | 0.615 ± 0.093 | 0.531 ± 0.049 | **0.328 ± 0.010** | **0.371 ± 0.007** |
| | 96 | 0.603 ± 0.051 | 0.581 ± 0.027 | 0.572 ± 0.039 | 0.553 ± 0.030 | 0.668 ± 0.031 | 0.555 ± 0.015 | **0.379 ± 0.011** | **0.406 ± 0.007** |
| | 288 | 0.849 ± 0.095 | 0.702 ± 0.051 | 0.789 ± 0.069 | 0.677 ± 0.039 | 0.795 ± 0.070 | 0.623 ± 0.032 | **0.451 ± 0.016** | **0.445 ± 0.008** |
| | 672 | 0.860 ± 0.057 | 0.726 ± 0.026 | 0.958 ± 0.183 | 0.758 ± 0.076 | 1.657 ± 1.116 | 0.890 ± 0.290 | **0.555 ± 0.011** | **0.511 ± 0.008** |
| | 1344 | 14.613 ± 26.108 | 1.948 ± 1.655 | 5.592 ± 7.032 | 1.497 ± 0.671 | - | - | **0.631 ± 0.061** | **0.556 ± 0.020** |
| ECL | 24 | 0.279 ± 0.007 | 0.372 ± 0.003 | 0.198 ± 0.005 | 0.310 ± 0.003 | 0.242 ± 0.005 | 0.332 ± 0.006 | **0.176 ± 0.002** | **0.285 ± 0.001** |
| | 48 | 0.309 ± 0.007 | 0.388 ± 0.004 | 0.245 ± 0.009 | 0.343 ± 0.007 | 0.275 ± 0.007 | 0.352 ± 0.006 | **0.194 ± 0.001** | **0.301 ± 0.001** |
| | 168 | 0.333 ± 0.016 | 0.410 ± 0.012 | 0.285 ± 0.006 | 0.375 ± 0.004 | - | - | **0.218 ± 0.002** | **0.320 ± 0.001** |
| | 336 | 0.326 ± 0.004 | 0.406 ± 0.001 | 0.304 ± 0.019 | 0.393 ± 0.013 | - | - | **0.241 ± 0.005** | **0.337 ± 0.002** |
| | 720 | 0.420 ± 0.094 | 0.467 ± 0.058 | 0.378 ± 0.083 | 0.443 ± 0.056 | - | - | **0.303 ± 0.012** | **0.383 ± 0.011** |
| | 960 | 0.399 ± 0.022 | 0.455 ± 0.017 | 0.360 ± 0.037 | 0.433 ± 0.027 | - | - | **0.325 ± 0.019** | **0.398 ± 0.015** |
| M4 | average | 0.224 ± 0.004 | 0.207 ± 0.003 | 0.223 ± 0.004 | 0.206 ± 0.003 | 0.223 ± 0.001 | 0.204 ± 0.001 | **0.208 ± 0.001** | **0.197 ± 0.001** |

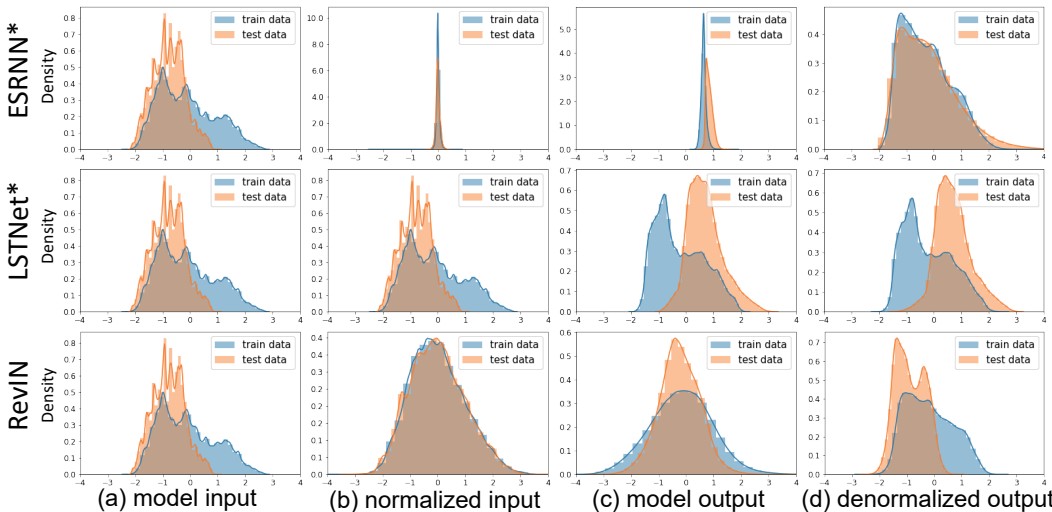

Figure 8: **Effect of RevIN on distribution discrepancy on training and test data compared to existing dynamic normalization methods.** We compare RevIN with LSTNet[*], which adds the autoregressive linear bypass module of LSTNet to the baseline and ES-RNN[*], which adds the exponential smoothing of ES-RNN to the baseline. The analysis is conducted on the ETTh$_2$ dataset with a prediction length of 960 using N-BEATS as the baseline. From left to right, the columns compare the training and test data distributions of each step of the sequential process in each method.

### A.6 COMPARISON WITH EXISTING DYNAMIC NORMALIZATION METHODS

We compare RevIN with the dynamic normalization methods proposed in LSTNet (Lai et al., 2017) and ES-RNN (Smyl, 2020). Similar to adding RevIN to the baseline model, we add the autoregressive linear bypass module of LSTNet and the modified Holt-Winters exponential smoothing of ES-RNN to the baseline model, respectively. As shown in Table 9, RevIN consistently achieves the best performance among the normalization methods adopted on N-BEATS by a significant margin. When we replace RevIN with the other normalization methods, the autoregressive linear bypass module of LSTNet (LSTNet[*]) also consistently reduces the prediction error compared to the baseline. However, the performance improvement is smaller than our method. For example, when the prediction length is 960 on the ETTh$_2$ dataset, N-BEATS shows an average error of 6.408, and LSTNet[*] reduces the error to 5.627. But this is still much worse than RevIN, which reduces the error to 0.471. Similarly, when the prediction length is 1344 on the ETTm$_1$ dataset, the baseline shows an average error of 14.613 and LSTNet[*] largely decreases the error to 5.592, but RevIN more significantly decreases the error to 0.631. In the case of ES-RNN[*], the training of the model is unstable, failing to converge for several cases. Also, ES-RNN[*] often degrades the baseline performance, e.g., when the prediction length is either 24 or 48 on the ETTh$_1$ and ETTm$_1$ datasets. It significantly reduces the error for long prediction length much better than LSTNet[*], but still worse than RevIN, for example, when the prediction length is 960 on the ETTh$_2$ dataset.

Additionally, we further analyze the data distributions of the dynamic normalization methods on the ETTh$_2$ dataset, as shown in Fig. 8. We compare the training and test data distributions of each step of the sequential process in each method.

**ES-RNN[*]** shows the distributions of (a) the original model input, (b) the normalized input where the level and seasonality are removed by its proposed method, (c) the model prediction output, (d) the denormalized output where the level and seasonality is multiplied back to the original distribution.

**LSTNet[*]** shows the distributions of (a) the original model input, (b) the same original input since the method does not transform the input data before feeding them to the main prediction model, the model prediction output (c) before, and (d) after adding the output of the autoregressive network.

**RevIN** shows the distributions of (a) the original model input, (b) the normalized input by RevIN, (c) the model prediction output, and (d) the denormalized output by RevIN.

In Fig. 8(a), the original training and test data show a discrepancy in their distributions. Also, they have several peaks, not being centered on the mean. This implies that sequences in the data will have different mean values. In Fig. 8(b), both RevIN and ES-RNN* transform data distributions into mean-centered distributions. Particularly, ES-RNN* extremely concentrates the distribution on the mean, with only a small variance. Both RevIN and ES-RNN* result in data sequences with similar statistics, thereby alleviating the distribution shift problem in the input data. This leads to outstanding performance on long prediction sequences, in contrast to LSTNet*. LSTNet* cannot resolve the distribution discrepancy because it does not have any module that can change the input statistics. Also, in LSTNet*, the distributions of the model output (Fig. 8(c)) completely differ from the input data distributions (Fig. 8(b)). The method cannot make the input and output distribution to be consistent. In addition, its proposed autoregressive model barely affects the model output distributions, as shown in Fig. 8(c-d); there is almost no difference between the model output and the final output distributions.

Most importantly, the distributions of the final output (Fig. 8(d)) significantly differ from the original data (Fig. 8(a)) in LSTNet*. Similarly, although ES-RNN* alleviates the distribution discrepancy in the input data, it fails to return the model output (Fig. 8(d)) back to the original distribution (Fig. 8(a)), especially with the test data. These results imply that LSTNet* and ES-RNN* fail to learn the appropriate data distribution, and this could be the main reason why their prediction error is higher than RevIN. On the other hand, in RevIN, the distributions of the final output (Fig. 8(d)) are successfully returned to the original distributions (Fig. 8(a)). Also, with RevIN, the input and output of the model maintain consistent distributions, as well as the training and test data be overlapped. As a result, RevIN shows superior performance than the other dynamic normalization methods.

Table 10: **Additional results on the comparison with classical and state-of-the-art normalization methods in Table 3 in the main manuscript.** The mean squared error is measured on the $ETTh_1$, $ETTh_2$, $ETTm_1$, and ECL datasets. $T_y$ indicates the prediction length. **RevBN** is the modified version of RevIN, where the input normalization is replaced by batch normalization.

| Dataset | $T_y$ | Min-max norm | z-score norm | Layer norm | DAIN | Batch norm | **RevBN** | Instance norm | **RevIN (Ours)** |
|---|---|---|---|---|---|---|---|---|---|
| $ETTh_1$ | 24 | 0.885 | 0.959 | 0.472 | 0.652 | 0.451 | 0.574 | 0.989 | **0.322** |
| | 48 | 1.010 | 0.898 | 0.741 | 1.389 | 0.557 | 0.649 | 0.999 | **0.373** |
| | 168 | 1.074 | 0.953 | 0.871 | 0.996 | 0.851 | 0.717 | 0.946 | **0.515** |
| | 336 | 1.083 | 0.969 | 0.827 | 0.979 | 0.828 | 0.775 | 1.078 | **0.509** |
| | 720 | 1.226 | 0.978 | 1.184 | 1.014 | 0.916 | 0.705 | 0.986 | **0.567** |
| | 960 | 1.224 | 1.043 | 1.303 | 1.032 | 1.691 | 0.779 | 1.090 | **0.697** |
| $ETTh_2$ | 24 | 2.659 | 3.152 | 0.478 | 1.437 | 0.336 | 0.550 | 2.976 | **0.192** |
| | 48 | 2.772 | 3.232 | 1.335 | 1.476 | 1.018 | 1.058 | 3.175 | **0.244** |
| | 168 | 2.987 | 3.329 | 4.092 | 1.982 | 6.206 | 0.729 | 3.240 | **0.419** |
| | 336 | 2.914 | 3.288 | 4.207 | 2.631 | 5.422 | 0.546 | 3.186 | **0.452** |
| | 720 | 3.092 | 3.031 | 5.822 | 2.954 | 7.062 | 1.552 | 3.079 | **0.492** |
| | 960 | 3.308 | 3.087 | 5.204 | 2.802 | 7.755 | 2.148 | 3.145 | **0.465** |
| $ETTm_1$ | 24 | 0.981 | 0.930 | 0.515 | 0.431 | 0.477 | 0.680 | 0.926 | **0.395** |
| | 48 | 0.998 | 1.005 | 0.555 | 0.747 | 0.489 | 0.531 | 1.005 | **0.337** |
| | 96 | 1.035 | 1.016 | 0.502 | 0.672 | 0.505 | 0.601 | 1.021 | **0.388** |
| | 288 | 0.974 | 0.988 | 0.773 | 0.877 | 0.677 | 0.656 | 1.056 | **0.444** |
| | 672 | 1.157 | 1.029 | 0.795 | 1.043 | 0.620 | 0.670 | 1.157 | **0.549** |
| | 1344 | 1.320 | 1.274 | 2.488 | 1.348 | 1.147 | 0.991 | 1.329 | **0.602** |
| ECL | 24 | 0.370 | 0.313 | 0.294 | 0.348 | 0.301 | 0.304 | 0.307 | **0.174** |
| | 48 | 0.334 | 0.326 | 0.310 | 0.387 | 0.319 | 0.331 | 0.313 | **0.194** |
| | 168 | 0.374 | 0.335 | 0.343 | 0.347 | 0.320 | 0.327 | 0.333 | **0.220** |
| | 336 | 0.378 | 0.338 | 0.358 | 0.357 | 0.337 | 0.369 | 0.335 | **0.244** |
| | 720 | 0.746 | 0.417 | 0.371 | 0.366 | 0.374 | 0.440 | 0.378 | **0.294** |
| | 960 | 0.387 | 0.377 | 0.379 | 0.381 | 0.422 | 0.414 | 0.386 | **0.329** |

## A.7 Additional Results on Comparison with Existing Normalization Methods

This section provides complete results that compare with existing normalization methods, which are not included in the main manuscript due to lack of space. The forecasting error of RevIN and existing normalization methods are evaluated on the ETTh$_1$, ETTh$_2$, ETTm$_1$, and ECL datasets in Table 10. RevIN consistently outperforms the other normalization methods across all datasets. Interestingly, when the denormalization step is added to batch normalization (RevBN) as RevIN, the model better forecasts long sequences than batch normalization (Batch norm), e.g., when the prediction length is 960. The denormalization step of RevIN plays a critical role in improving model performance by restoring the model prediction to the original distribution. However, a denormalization step cannot be added to DAIN since it has the Hadamard multiplication operation in the last step, which is not reversible when the input and prediction sequence lengths are different. These differences could be the reason for its worse performance compared to RevIN despite that DAIN requires higher computational costs and a larger amount of model parameters.

## A.8 Algorithm of Reversible Instance Normalization

Algorithm 1 summarizes the procedure of the proposed approach. Reversible instance normalization consists of the normalization (line 3-4) and denormalization layers (line 6-7). It transforms the input and output of a model using identical statistics. As RevIN is generally applicable, $g_\theta$ in Algorithm 1 (line 5) can be any arbitrary deep neural network.

---

**Algorithm 1:** RevIN, applied to input $x$ and output $y$ of a module in the model.

---

**Input** : $T_x \in \mathbb{R}^1$, the input sequence length; $x_{kt}^{(i)} \in \mathbb{R}^1$, the $k$-th feature at time step $t$ of the $i$-th item in a mini-batch; $\gamma, \beta \in \mathbb{R}^K$, learnable parameters for RevIN; $g_\theta$, a module in the model parameterized by $\theta$.

**Output:** $\gamma, \beta, \theta$.

1    *Compute*    $\mu_\mathcal{T} \leftarrow \frac{1}{T_x} \sum_{j=1}^{T_x} x_{kj}^{(i)}$             ▷ instance mean

2    *Compute*    $\sigma_\mathcal{T}^2 \leftarrow \frac{1}{T_x} \sum_{j=1}^{T_x} (x_{kj}^{(i)} - \mu_\mathcal{T})^2$          ▷ instance variance

3    *Normalize*    $\hat{x}_{kt}^{(i)} \leftarrow \frac{x_{kt}^{(i)} - \mu_\mathcal{T}}{\sqrt{\sigma_\mathcal{T}^2 + \epsilon}}$            ▷ normalization

4    *Transform*    $\hat{x}_{kt}^{(i)} \leftarrow \gamma_k \cdot \hat{x}_{kt}^{(i)} + \beta_k \equiv \mathbf{RevIN}_{\gamma,\beta}^{\mathrm{n}}(x_{kt}^{(i)})$      ▷ scale and shift

5    *Predict*    $\tilde{y} \leftarrow g_\theta(\hat{x})$                  ▷ forward propagation

6    *Retransform*    $\hat{y}_{kt}^{(i)} \leftarrow \frac{\tilde{y}_{kt}^{(i)} - \beta_k}{\gamma_k}$           ▷ reverse scale and shift

7    *Denormalize*    $\hat{y}_{kt}^{(i)} \leftarrow \mu_\mathcal{T} + \hat{y}_{kt}^{(i)} \sqrt{\sigma_\mathcal{T}^2 + \epsilon} \equiv \mathbf{RevIN}_{\gamma,\beta}^{\mathrm{dn}}(\tilde{y}_{kt}^{(i)})$    ▷ denormalization

---

## A.9 Theoretical Justification of RevIN Against Distribution Shift

Let $x^{(i)} \in \mathbb{R}^{K \times T_x}$ denote a time series comprising $K$ variables of length $T_x$. Consider a univariate case where $K = 1$ without the loss of generality. Then, $x^{(i)} \in \mathbb{R}^{T_x}$ denotes the $i$-th time series in the data. Consider training and test data, whose distributions are denoted as $P_{tra}$ and $P_{tst}$, respectively. We consider a distribution shift problem where the training and test data have different distributions (Du et al., 2021). That is,

$$P_{tra}(x) \neq P_{tst}(x). \tag{4}$$

In our work, we consider the distribution shift problem in terms of the mean and the variance. Then, the distribution shift problem can be redefined as

$$\mathbb{E}[x_{tra}] \neq \mathbb{E}[x_{tst}] \ \text{ or } \ \mathrm{Var}[x_{tra}] \neq \mathrm{Var}[x_{tst}], \ \text{ for } x_{tra} \sim P_{tra}, \ x_{tst} \sim P_{tst}. \tag{5}$$

Let's assume that the given training and test data suffer from the distribution shift problem in terms of the mean and variance. In order to solve this problem, RevIN first normalizes a training sample

$x^{(i)} \sim P_{tra}$. Mathematically, a training sample is transformed as

$$\hat{x}^{(i)} = \frac{x^{(i)} - \mathbb{E}[x^{(i)}]}{\sqrt{\mathrm{Var}[x^{(i)}]}} \cdot \gamma + \beta. \tag{6}$$

By the laws of expectation and variance,

$$\mathbb{E}[\hat{x}^{(i)}] = \beta \ \text{ and } \ \mathrm{Var}[\hat{x}^{(i)}] = \gamma^2, \ \text{ for all } \ i. \tag{7}$$

By symmetry, this also holds for $x^{(i)} \sim P_{tst}$, for all $i$. Therefore, the mean and variance of the training and test data distributions become identical. Thus, by definition (Eq. 5), the distribution shift problem for the training and test data is solved by the first step of RevIN.

Given the normalized time-series data, the forecasting model parameterized by $\theta$, $f_\theta : \mathbb{R}^{T_x} \to \mathbb{R}^{T_y}$, predicts the corresponding subsequent future values, $\tilde{y} = f_\theta(\hat{x})$. Then, the denormalization step of RevIN returns the non-stationary information of the original data, i.e., $\mathbb{E}[x^{(i)}]$ and $\mathrm{Var}[x^{(i)}]$ in Eq. 6, to the prediction output so that model does not have to reconstruct them from the normalized input. In summary, the model prediction $\tilde{y}^{(i)}$ is denormalized as

$$\hat{y}^{(i)} = \frac{\tilde{y}^{(i)} - \beta}{\gamma} \cdot \sqrt{\mathrm{Var}[x^{(i)}]} + \mathbb{E}[x^{(i)}]. \tag{8}$$

By the laws of expectation and variance, the mean and variance of $\hat{y}^{(i)}$ can be expressed as

$$\mathbb{E}[\hat{y}^{(i)}] = \Delta + \mathbb{E}[x^{(i)}] \ \text{ and } \ \mathrm{Var}[\hat{y}^{(i)}] = \lambda \cdot \mathrm{Var}[x^{(i)}]. \tag{9}$$

The denormalization step allows the mean and variance of the final prediction values to be expressed as the difference from the input statistics. Here, since the input data $x^{(i)}$ and the groundtruth future values are consecutive sequences, we can assume that their difference in the mean and variance can be expressed as Eq. 9 as well. Under this assumption, the model adopting RevIN only needs to capture the difference from the input statistics, $\Delta$ and $\lambda$, to accurately predict the statistics of the future values. In conclusion, through the normalization and denormalization steps of RevIN, a model can focus on learning the offset from the input distribution to the output distribution by removing their common non-stationary statistics.

## A.10 Calculation Details on Feature Divergence

This section explains how the feature divergence is computed in Section 4.2.3. Following the previous work (Pan et al., 2018), we calculate the average feature divergence between the training and test data using symmetric KL divergence, assuming that the output features of the model layer will follow a Gaussian distribution with mean $\mu$ and variance $\sigma^2$. Then, the equation for the feature divergence of the $k$-th feature $f_k$ can be expressed as

$$D(f_k^{\text{train}} || f_k^{\text{test}}) = KL(f_k^{\text{train}} || f_k^{\text{test}}) + KL(f_k^{\text{test}} || f_k^{\text{train}}), \tag{10}$$

$$\text{where} \quad KL(f_k^A || f_k^B) = \log \frac{\sigma_k^B}{\sigma_k^A} + \frac{\sigma_k^{A2} + (\mu_k^A - \mu_k^B)^2}{2\sigma_k^{B2}} - \frac{1}{2}. \tag{11}$$

## A.11 Additional experimental Details

We train and evaluate the models using the following seeds: 12, 22, 32, 42, and 52. The experiments using N-BEATS and Informer as the baseline are performed on NVIDIA TITAN RTX, and the experiments using SCINet are conducted on NVIDIA TITAN Xp. Following the multivariate time-series forecasting settings of the previous studies (Zhou et al., 2021; Liu et al., 2021), we select input sequence length from two days (2d), 4d, 7d, 14d, 15d, 20d, 28d, 30d for the hourly datasets, i.e., the ETTh$_1$, ETTh$_2$, and ECL datasets, and from half day, 1d, 3.5d, 7d for the ETTm$_1$ dataset. Particularly, we set the ratio of input length to prediction length to be smaller from 2.0 to 0.35 as the prediction length becomes longer. In the case of SCINet, when the prediction length is 720, we set its input length to be 736, unlike the other baselines. It is because the original paper of the method requires its input sequence length to meet a specific condition. To be specific, the input length needs to be a multiple of 32 due to its hierarchical architecture (Liu et al., 2021).

A.12    REPRODUCTION DETAILS FOR BASELINE MODELS

This section describes the implementation details of the baselines, Informer, N-BEATS, and SCINet. Note that we compare each baseline model and RevIN using the same hyperparameters except for the presence of RevIN. We exactly follow the experimental settings of the baseline models by using their official code to conduct experiments, except for N-BEATS that have no officially released code. We reproduce the model and set hyperparameters as stated in the original N-BEATS paper.

**Informer.** We use the official open-source code of Informer [6]. If provided, we follow the same hyperparameter settings in training the network, e.g., hidden dimension of the network or the learning rate. For the ECL dataset, detailed hyperparameter settings are not officially provided in Informer; we use the same settings with the ETTh$_1$ dataset.

**N-BEATS.** We reproduce N-BEATS using the PyTorch framework (Paszke et al., 2019). We follow the same hyperparameter settings of the N-BEATS-I model in the original paper. We train N-BEATS to minimize the mean squared error between the model prediction and groundtruth values. For a fair comparison with the other baselines, we use a single model instead of using the ensemble method originally proposed in the N-BEATS paper. Since N-BEATS is a model tailored to univariate time-series forecasting, we flatten each multivariate input sequence into a univariate sequence having a single dimension for the feature before feeding it to the model. Additionally, we conduct a grid search for the learning rate of N-BEATS with the range of [1e-5, 1e-3] and train the model using the weight decay with the factor of 0.001 to stabilize training.

**SCINet.** We follow the experimental settings provided in the official code [7] of SCINet.

A.13    ADDITIONAL QUALITATIVE RESULTS

In Fig. 9, we illustrate the additional results comparing the predictions of RevIN and the baselines. Overall, the prediction results of the baselines are inaccurately scaled and shifted. However, RevIN shows remarkable performance, consistently improving the baselines to predict more precise results. With RevIN, the forecasting results are better aligned with the groundtruth.

A.14    COMPLETE QUANTITATIVE RESULTS

Table 11 provides the standard deviation values of five independent experiments to compare long sequence forecasting performance in Table 2 in the main manuscript. Table 12 shows the complete results of the comparison of the forecasting errors between the baselines and RevIN in Table 1 in the main manuscript. They include the standard deviation of five runs and the performance reported in the original papers of the baselines. RevIN shows significant performance improvement compared to the state-of-the-art forecasting baselines.

Table 11: **Standard deviation values of the five runs for the comparison of long sequence forecasting performance in Table 2 in the main manuscript.**

| Prediction length | 48 | | 168 | | 336 | | 720 | | 960 | |
|---|---|---|---|---|---|---|---|---|---|---|
| Metric | MSE | MAE | MSE | MAE | MSE | MAE | MSE | MAE | MSE | MAE |
| Informer | 0.056 | 0.035 | 0.052 | 0.024 | 0.085 | 0.031 | 0.037 | 0.024 | 0.034 | 0.022 |
| **+ RevIN** | 0.030 | 0.008 | 0.051 | 0.030 | 0.073 | 0.026 | 0.067 | 0.033 | 0.041 | 0.022 |
| N-BEATS | 0.042 | 0.030 | 0.056 | 0.027 | 0.081 | 0.037 | 0.072 | 0.029 | 0.067 | 0.030 |
| **+ RevIN** | 0.006 | 0.003 | 0.014 | 0.007 | 0.011 | 0.006 | 0.041 | 0.020 | 0.027 | 0.012 |
| SCINet | 0.008 | 0.007 | 0.063 | 0.042 | 0.115 | 0.064 | 0.033 | 0.022 | 0.049 | 0.029 |
| **+ RevIN** | 0.002 | 0.001 | 0.013 | 0.007 | 0.022 | 0.010 | 0.028 | 0.017 | 0.015 | 0.008 |

---

[6]https://github.com/zhouhaoyi/Informer2020
[7]https://github.com/cure-lab/SCINet

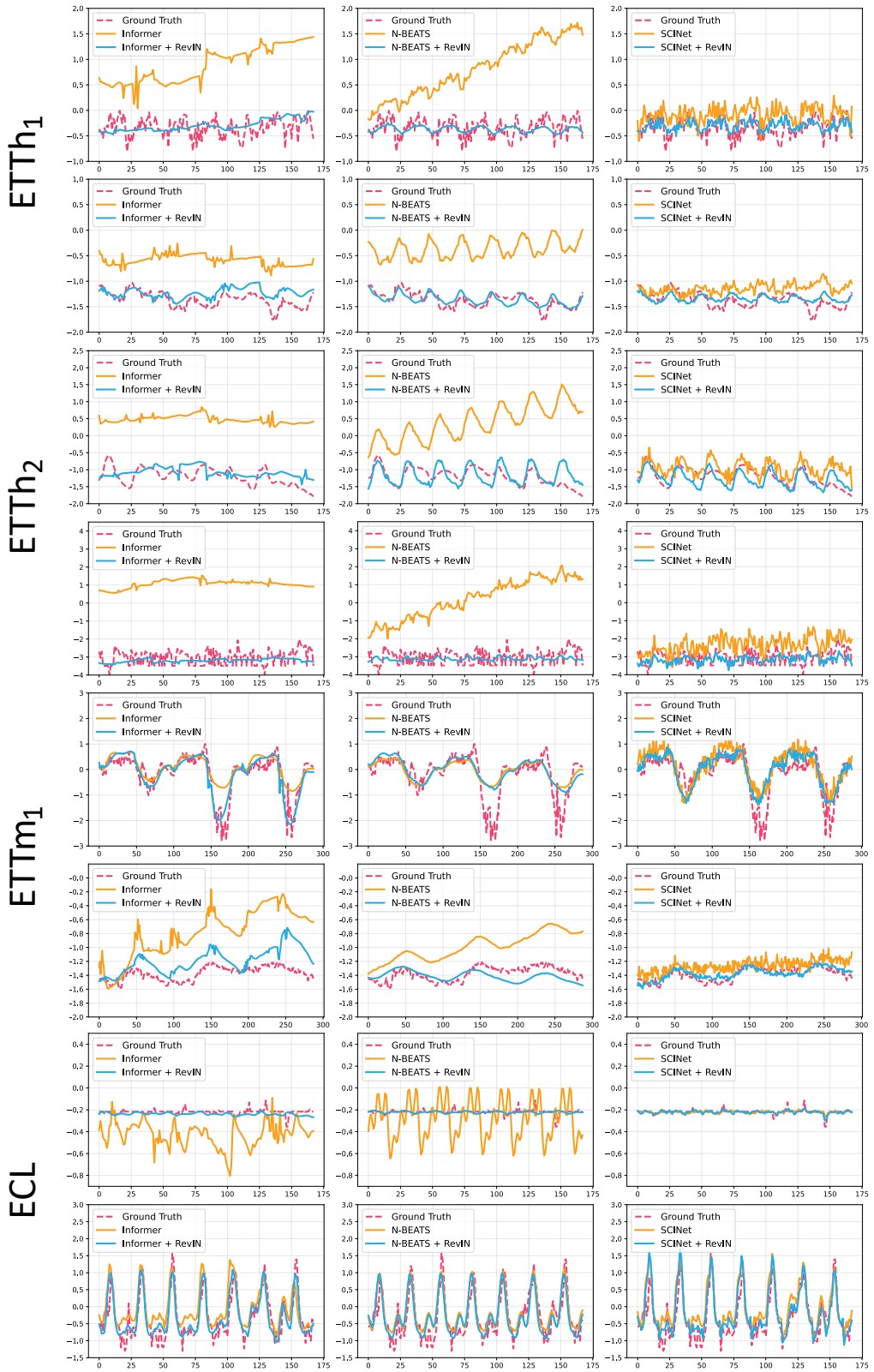

Figure 9: **Additional multivariate time-series forecasting results comparing RevIN and state-of-the-art baselines.** The analysis is conducted on the $ETTh_1$, $ETTh_2$, $ETTm_1$, and ECL datasets. We set the prediction length as 168 for the hour datasets and 288 for the $ETTm_1$ dataset.

Table 12: **Complete results for the comparison of the forecasting errors in Table 1 in the main manuscript.** The mean and standard deviation of the five independent experiments are recorded. † denotes the reported performances for the baselines in their original paper.

| Dataset | Metric | Informer† | | Informer | | Informer +RevIN | | N-BEATS | | N-BEATS+RevIN | | SCINet† | | SCINet | | SCINet+RevIN | |
|---|---|---|---|---|---|---|---|---|---|---|---|---|---|---|---|---|---|
| | | MSE | MAE | MSE | MAE | MSE | MAE | MSE | MAE | MSE | MAE | MSE | MAE | MSE | MAE | MSE | MAE |
| ETTh1 | 24 | 0.577 | 0.549 | 0.550 ±0.041 | 0.536 ±0.025 | 0.504 ±0.049 | 0.472 ±0.025 | 0.478 ±0.022 | 0.505 ±0.012 | 0.330 ±0.006 | 0.373 ±0.004 | 0.311 | 0.348 | 0.338 ±0.012 | 0.373 ±0.009 | 0.308 ±0.003 | 0.347 ±0.002 |
| | 48 | 0.685 | 0.625 | 0.772 ±0.122 | 0.668 ±0.055 | 0.646 ±0.039 | 0.547 ±0.015 | 0.536 ±0.041 | 0.542 ±0.012 | 0.372 ±0.001 | 0.400 ±0.002 | 0.364 | 0.388 | 0.436 ±0.025 | 0.459 ±0.021 | 0.365 ±0.005 | 0.389 ±0.003 |
| | 168 | 0.931 | 0.752 | 1.138 ±0.096 | 0.853 ±0.045 | 0.655 ±0.055 | 0.561 ±0.024 | 1.005 ±0.060 | 0.782 ±0.041 | 0.466 ±0.001 | 0.452 ±0.002 | 0.497 | 0.491 | 0.459 ±0.015 | 0.461 ±0.013 | 0.406 ±0.003 | 0.416 ±0.003 |
| | 336 | 1.128 | 0.873 | 1.278 ±0.129 | 0.909 ±0.058 | 1.058 ±0.119 | 0.758 ±0.059 | 0.932 ±0.146 | 0.743 ±0.064 | 0.515 ±0.013 | 0.483 ±0.008 | 0.491 | 0.494 | 0.527 ±0.010 | 0.513 ±0.006 | 0.467 ±0.005 | 0.471 ±0.003 |
| | 720 | 1.215 | 0.896 | 1.357 ±0.056 | 0.945 ±0.009 | 0.926 ±0.057 | 0.717 ±0.036 | 1.389 ±0.079 | 0.926 ±0.042 | 0.576 ±0.035 | 0.534 ±0.018 | 0.612 | 0.582 | 0.596 ±0.015 | 0.571 ±0.013 | 0.507 ±0.006 | 0.505 ±0.004 |
| | 960 | . | . | 1.470 ±0.124 | 0.990 ±0.052 | 0.902 ±0.033 | 0.715 ±0.025 | 1.383 ±0.380 | 0.932 ±0.120 | 0.678 ±0.019 | 0.575 ±0.009 | . | . | 0.604 ±0.017 | 0.574 ±0.014 | 0.545 ±0.010 | 0.526 ±0.005 |
| ETTh2 | 24 | 0.720 | 0.665 | 0.450 ±0.099 | 0.520 ±0.071 | 0.238 ±0.010 | 0.325 ±0.006 | 0.403 ±0.185 | 0.472 ±0.101 | 0.192 ±0.003 | 0.276 ±0.002 | 0.183 | 0.271 | 0.199 ±0.026 | 0.295 ±0.027 | 0.180 ±0.004 | 0.263 ±0.002 |
| | 48 | 1.457 | 1.001 | 2.171 ±0.094 | 1.200 ±0.048 | 0.361 ±0.023 | 0.404 ±0.014 | 1.330 ±0.240 | 0.918 ±0.073 | 0.254 ±0.011 | 0.320 ±0.008 | 0.259 | 0.341 | 0.350 ±0.025 | 0.422 ±0.027 | 0.231 ±0.006 | 0.302 ±0.006 |
| | 168 | 3.489 | 1.515 | 8.157 ±0.631 | 2.558 ±0.113 | 0.859 ±0.072 | 0.649 ±0.026 | 7.174 ±0.449 | 2.329 ±0.049 | 0.410 ±0.010 | 0.418 ±0.005 | 0.528 | 0.509 | 0.559 ±0.044 | 0.518 ±0.025 | 0.337 ±0.007 | 0.378 ±0.003 |
| | 336 | 2.723 | 1.340 | 4.746 ±0.455 | 1.844 ±0.102 | 0.890 ±0.057 | 0.673 ±0.023 | 4.859 ±0.268 | 1.863 ±0.043 | 0.449 ±0.011 | 0.447 ±0.006 | 0.648 | 0.608 | 0.664 ±0.073 | 0.583 ±0.030 | 0.357 ±0.003 | 0.403 ±0.002 |
| | 720 | 3.467 | 1.473 | 3.190 ±0.326 | 1.529 ±0.085 | 0.576 ±0.044 | 0.546 ±0.025 | 5.656 ±1.053 | 2.012 ±0.186 | 0.496 ±0.008 | 0.482 ±0.002 | 1.074 | 0.761 | 1.546 ±0.378 | 0.944 ±0.141 | 0.411 ±0.003 | 0.445 ±0.002 |
| | 960 | . | . | 2.972 ±0.183 | 1.441 ±0.035 | 0.600 ±0.033 | 0.570 ±0.018 | 6.408 ±2.039 | 2.077 ±0.242 | 0.471 ±0.015 | 0.481 ±0.008 | . | . | 1.862 ±0.153 | 1.066 ±0.055 | 0.438 ±0.007 | 0.462 ±0.004 |
| ETTm1 | 24 | 0.323 | 0.369 | 0.330 ±0.021 | 0.382 ±0.017 | 0.309 ±0.020 | 0.352 ±0.010 | 0.443 ±0.043 | 0.437 ±0.035 | 0.403 ±0.006 | 0.392 ±0.005 | 0.127 | 0.226 | 0.130 ±0.003 | 0.231 ±0.003 | 0.106 ±0.002 | 0.196 ±0.001 |
| | 48 | 0.494 | 0.503 | 0.499 ±0.024 | 0.486 ±0.012 | 0.390 ±0.008 | 0.391 ±0.006 | 0.453 ±0.034 | 0.472 ±0.018 | 0.328 ±0.010 | 0.371 ±0.007 | 0.150 | 0.261 | 0.155 ±0.004 | 0.262 ±0.004 | 0.135 ±0.003 | 0.222 ±0.002 |
| | 96 | 0.678 | 0.614 | 0.605 ±0.033 | 0.554 ±0.027 | 0.405 ±0.013 | 0.411 ±0.006 | 0.603 ±0.051 | 0.581 ±0.027 | 0.379 ±0.011 | 0.406 ±0.007 | 0.190 | 0.291 | 0.195 ±0.012 | 0.291 ±0.013 | 0.162 ±0.001 | 0.247 ±0.001 |
| | 288 | 1.056 | 0.786 | 0.906 ±0.039 | 0.738 ±0.028 | 0.563 ±0.024 | 0.502 ±0.015 | 0.849 ±0.095 | 0.702 ±0.051 | 0.451 ±0.016 | 0.445 ±0.008 | 0.417 | 0.462 | 0.361 ±0.008 | 0.419 ±0.004 | 0.265 ±0.003 | 0.321 ±0.002 |
| | 672 | 1.192 | 0.926 | 0.943 ±0.062 | 0.760 ±0.034 | 0.663 ±0.082 | 0.550 ±0.031 | 0.860 ±0.057 | 0.726 ±0.026 | 0.555 ±0.011 | 0.511 ±0.008 | 0.554 | 0.527 | 1.020 ±0.040 | 0.756 ±0.025 | 0.357 ±0.004 | 0.380 ±0.002 |
| | 1344 | . | . | 1.095 ±0.065 | 0.823 ±0.040 | 0.824 ±0.039 | 0.632 ±0.019 | 14.613 ±26.108 | 1.948 ±1.655 | 0.631 ±0.061 | 0.556 ±0.020 | . | . | 1.841 ±0.242 | 1.044 ±0.100 | 0.412 ±0.008 | 0.422 ±0.003 |
| ECL | 24 | . | . | 0.250 ±0.005 | 0.358 ±0.005 | 0.148 ±0.001 | 0.257 ±0.001 | 0.279 ±0.007 | 0.372 ±0.003 | 0.176 ±0.002 | 0.285 ±0.001 | . | . | 0.138 ±0.004 | 0.246 ±0.005 | 0.112 ±0.001 | 0.207 ±0.001 |
| | 48 | 0.344 | 0.393 | 0.300 ±0.010 | 0.386 ±0.005 | 0.171 ±0.004 | 0.279 ±0.003 | 0.309 ±0.007 | 0.388 ±0.004 | 0.194 ±0.001 | 0.301 ±0.001 | . | . | 0.163 ±0.007 | 0.265 ±0.007 | 0.126 ±0.001 | 0.222 ±0.001 |
| | 168 | 0.368 | 0.424 | 0.345 ±0.012 | 0.423 ±0.010 | 0.261 ±0.010 | 0.354 ±0.007 | 0.333 ±0.016 | 0.410 ±0.012 | 0.218 ±0.002 | 0.320 ±0.001 | . | . | 0.177 ±0.003 | 0.281 ±0.005 | 0.153 ±0.003 | 0.249 ±0.002 |
| | 336 | 0.381 | 0.431 | 0.429 ±0.041 | 0.473 ±0.023 | 0.356 ±0.026 | 0.414 ±0.015 | 0.326 ±0.004 | 0.406 ±0.001 | 0.241 ±0.005 | 0.337 ±0.002 | . | . | 0.202 ±0.004 | 0.308 ±0.004 | 0.162 ±0.001 | 0.262 ±0.001 |
| | 720 | 0.406 | 0.443 | 0.851 ±0.088 | 0.719 ±0.072 | 0.834 ±0.122 | 0.700 ±0.076 | 0.420 ±0.094 | 0.467 ±0.058 | 0.303 ±0.012 | 0.383 ±0.011 | . | . | 0.234 ±0.006 | 0.333 ±0.004 | 0.183 ±0.003 | 0.281 ±0.002 |
| | 960 | 0.460 | 0.548 | 0.930 ±0.075 | 0.750 ±0.042 | 0.894 ±0.047 | 0.741 ±0.031 | 0.399 ±0.022 | 0.455 ±0.017 | 0.325 ±0.019 | 0.398 ±0.015 | . | . | 0.235 ±0.011 | 0.330 ±0.008 | 0.200 ±0.003 | 0.292 ±0.002 |

