# OpenReview forum: "Reversible Instance Normalization for Accurate Time-Series Forecasting against Distribution Shift"
_ICLR.cc/2022/Conference — ICLR 2022 Poster_

### Official Review · Reviewer_J9jm · 2021-11-02

**Correctness:** 3
**Technical Novelty And Significance:** 2
**Empirical Novelty And Significance:** 2
**Recommendation:** 5
**Confidence:** 4

**Main Review:**

Strengths:
The idea behind RevIN is straightforward but seems to work well, outperforming min-max normalization, batch normalization, layer normalization and instance normalization. RevIN is also cheap to compute and model-agnostic.

Weaknesses:
W1: The method seems ad-hoc with no theoretical justification. Is is possible to provide a proof for a simple case (e.g. 1-dimensional time series) that RevIN indeed handles distribution shift over time?

W2: Experiments are on 4 data sets only. To validate the merit of RevIN, I would like to see experimental results on more real-world data sets, for instance those taken from UCI Repository.

Other remarks:
R1: In the first paragraph of Section 3.1, it looks like output sequence $Y$ needs to have the same length as the input sequence $X$. Does this have to be the case?

R2: ECL data set originally has a lot of dimensions and a few instances. After preprocessing, what are the length and dimensionality of the new time series?

**Summary Of The Paper:**

The paper proposes RevIN that performs (a) instance normalization on input time series and (b) reversible instance normalization on output time series using normalization statistics of the input. By doing this, the paper claims to handle distribution shift in time series and hence yield better forecasting results.

**Summary Of The Review:**

The paper is easy to follow as the main idea is straightforward. The technical contribution however seems ad-hoc and incremental. A proof for a simple case would make the paper more convincing.

---

> ### Author Response · Authors · 2021-11-17
> **Response to Reviewer J9jm**
>
> We appreciate your review and the positive comments regarding our paper. We would like to respond to your comments and hope that we address all your concerns below:
>
> **Q1. The method seems ad-hoc with no theoretical justification. Is is possible to provide a proof for a simple case (e.g. 1-dimensional time series) that RevIN indeed handles distribution shift over time?**
>
> A1. We add the theoretical justification that RevIN reduces the distribution shift problem in time-series in **Appendix A.9** in our manuscript. Note that we also have shown on the real-world data that RevIN alleviates the discrepancy in train and test data distributions (thereby solving the distribution shift problem) in Fig. 3.
>
> **Q2. Experiments are on 4 data sets only. To validate the merit of RevIN, I would like to see experimental results on more real-world data sets, for instance those taken from UCI Repository.**
>
> A2. Please see the general response, titled **“General Response: Experimental Results on Additional Real-World Datasets.”**
>
> **Q3. Other remarks:**
> 1) **In the first paragraph of Section 3.1, it looks like output sequence $Y$ needs to have the same length as the input sequence $X$. Does this have to be the case?**
> 2) **ECL data set originally has a lot of dimensions and a few instances. After preprocessing, what are the length and dimensionality of the new time series?**
>
> A3.
> 1) In RevIN, the input length $T_x$ and the prediction length $T_y$ can be different since observations are normalized (and denormalized) across the temporal dimension. Considering the $i$-th item in a mini-batch, $x^{(i)} \in \mathbb{R}^{(K,   T_x)}$, where $K$ denotes the number of variables, the mean and standard deviation are computed on the dimension for $T_x$ as $\mu=\frac{1}{T_x}\sum_{i=1}^{T_x}x_i \in \mathbb{R}^{K}, \sigma^2=\frac{1}{T_x}\sum_{i=1}^{T_x}(x_i-\mu)^2\in \mathbb{R}^{K}$, respectively. Then, they are used to normalize the input $x^{(i)} \in \mathbb{R}^{(K,   T_x)}$ and denormalize the corresponding model prediction $y^{(i)} \in \mathbb{R}^{(K,   T_y)}$.
>
> 2) The original data have 322 variables with a length of 26303. Consider the case when the input length is 336, and the prediction length is 960.
> After applying the sliding window technique, the new data have 17117 different 321-dimensional sequences of length 336 for the input training data. The corresponding prediction targets for each input sequence are 17117 different 321-dimensional sequences of length 960. One of the variables indicates the measured time so it is excluded after the preprocessing (from 322 to 321).
> Note that the input and the prediction length can be different from each other. Likewise, the validation and test data have 1673 and 4301 sequences, respectively. In the case of the normalization step of RevIN, it does not change the length and dimensionality of the data.
> |                                    | split           | input data size (N, $T_x$, K)              | output data size (N, $T_y$, K) |
> |:------------------------------------|:-----------------|:-----------------------|:-----------------------|
> | **original data**                      | -               | (1, 26303, 322)       |-
> | **after sliding window**  | training data | (17117, 336, 321) | (17117, 960, 321)
> |                                | validation data | (1673, 336, 321)  | (1673, 960, 321)
> |                                    | test data       | (4301, 336, 321)  | (4301, 960, 321)

---

### Official Review · Reviewer_V5fo · 2021-11-02

**Correctness:** 3
**Technical Novelty And Significance:** 2
**Empirical Novelty And Significance:** Not applicable
**Recommendation:** 8
**Confidence:** 4

**Main Review:**

Some strengths:
* The approach is interesting and well motivated, and the experiment results useful, striking, and very extensive.
* It is a good data point to have to see the benefit of this type of normalization across modern state-of-the-art models, and point out this method in the community, which has typically just used global per-series normalization.
* The results conclusively show the benefit and the method provides a simple modification that can be broadly used in many cases, and the detailed results in the appendix that also include original reported scores and std. dev. are really great to have.
* The thorough comparisons to different normalization approaches are also helpful.
* Also the appendix details and code provided are very helpful for reproducibility


Weaknesses:
1. Novelty:

However, the biggest issue is that the proposed work is not very novel, and leaves out the related work on "sliding window normalization" doing essentially the same thing.  I.e., "sliding window normalization" is commonly used in practice for time series forecasting and has been used in the literature on time series forecasting as well.  Sliding window normalization typically just takes the current instance time window mean and std. dev. to normalize the inputs and de-normalize the outputs after getting the predictions.  For example, see the following, and references:

Ogasawara, Eduardo, et al. "Adaptive normalization: A novel data normalization approach for non-stationary time series." The 2010 International Joint Conference on Neural Networks (IJCNN). IEEE, 2010.

Overall it is a simple and straight-forward method, and, the only difference from the classical sliding window normalization approach is the additional learnable parameters (gamma and beta).  However, such learnable parameters are not new on their own either and already used in other general normalization approaches as well as forecasting specific - including instance normalization for forecasting, like DAIN (albeit in that case the difference is it doesn't use the built-in denormalization).

I think the key main missing experiment and result needed is to demonstrate the impact of the learnable parameters, gamma and beta.  I.e., in each of the experiments, if the learnable parameters are excluded what is the impact?  This amounts to an additional ablation study - the same proposed method is used, but with gamma and beta excluded (or fixed to 1 and 0 respectively so they have no effect).

Additionally, I think the hybrid class of models are very similar conceptually and should ideally be compared with and called out as an alternative approach to dynamic normalization.  I.e., the hybrid models fit a simple model essentially based on the recent history of each series combined with a more complex deep learning model - and one of the key motivations for this was to dynamically normalize each series using this simple model.  For example the author of ES-RNN explicitly said that the idea of using the exponential smoothing model was to enable per-series normalization before applying the deep learning part of the model - and it fits an exponential smoothing model to set the level and seasonal pattern (and this is inherently based on limited recent history which is controlled by the smoothing parameters that are learned as part of modeling).  Similarly LSTNet and other methods before and after it simultaneously fit a linear auto regressive model and add it to the neural net part of the model to get the final output.  Both of these kind of approaches are essentially normalizing the time series for the deep learning part, as is done here, and this is done dynamically and potentially capturing nonstationarity of the model as well in the normalization, which isn't done with the proposed approach.
ES-RNN is Smyl 2020 in the cited references and LSTNet is: Lai, Guokun, et al. "Modeling long-and short-term temporal patterns with deep neural networks." SIGIR 2018.
It would be best to compare with these (including the distributions) / compare this approach in combination with the baseline models as well - e.g., simultaneously fitting a simple baseline AR model + or * the deep learning model to get the output, as this is an alternate way to perform this kind of dynamic normalization, so it would be important to see how it compares to the proposed one.


2.
It seems a key hyper parameter is the window length, as clearly if the window length is too short, the instance normalization would hardly be stable (and enable accurate prediction with denormalization to the full horizon).  However, this is not discussed and related experiments not performed.

I.e., it would therefore be best to show the impact of changing window lengths - i.e., hyper parameter sensitivity as this is varied from a very short window to a longer and longer window (additionally at some point too long windows would also be problematic as it would not be adaptive enough and also limit the number of data points)

It's also important to make clear how the window length was selected for the experiments.


3. Additionally issues / open questions around the formulation and the experiments:
* The mathematical formulation is not precise / changes at different places. E.g., at first instances are referred to as x^(i), then suddenly referred to by x_k.
* Alg. 1 definition directly contradicts the description of the method and definitions in the main paper.  Namely, the main paper states that there are N time windows, each with K variables / time series (in the case of multivariate series), and that the scaling and shift parameters, gamma and beta, are in R^K, i.e., one shared value for each variable / dimension K - shared across input windows.  However, in Alg. 1 they are said to be in R^NxK - which implies a unique value per window and feature/variable is learned, which doesn't make sense (as then one wouldn't be able to apply it for future forecasts, e.g., series N+1 where we don't have a parameter).
* It would be best to make clear the dimensions of the different parameters, the dimensions of the inputs and outputs, and data
* It is also not explained how the hyper-parameters tuned, and it would be
* It would also best to show the results on a larger variety of datasets, even those used in prior work by the base methods - as the benefit might only be for this common type of data used here.



**Summary Of The Paper:**

The authors propose a "reversible instance normalization" as an input pre and post processing procedure to improve the forecasting of any given base model - targeted at addressing the distribution shift that is common in time series data - e.g., time series are typically non-stationary.

This works by normalizing each time window input to a (deep learning) forecast model, applying the model on the normalized data, then unnormalizing the predictions to get the final predictions.   The normalization is done by subtracting the mean and dividing by the std. dev. in a current (input instance) window, followed by scaling and shifting by learnable, shared cross-instance (global), scaling and shifting parameters per input feature / variable.

The authors perform extensive experiments to show the proposed approach significantly improves the base metric score results for 3 recent, state-of-the-art forecasting algorithms across several datasets, and further that it significantly improves over other normalization approaches, and helps align distributions between train and test windows.

**Summary Of The Review:**

My main hesitance to accept the paper is the lack of novelty - as sliding window normalization is already commonly used, and the method is a very minor change. I feel this could possibly be improved with the ablation study comparing the results of the proposed method without the learnable parameters gamma and beta.

I would also like to see more analyses and discussion around the impact of the window length used, and fixes / additional details around method and experiment details.   It would also be best to see the performance on a larger variety of datasets - in case the method is only beneficial for the type of data used here - and there are more datasets from the base method papers that could be used for this, that are left out here.  Additionally it would also be useful to see the results of a hybrid approach with the baselines as a commonly used alternative to the proposed dynamic normalization approach (e.g., fit a linear AR model + the baseline model).

Overall, I feel the paper provides an extensive study showcasing the benefit of the proposed normalization approach / sliding window normalization so it also provides value to the community even if lack of novelty - and highlights this approach to the community as an alternative to the global normalization that has become the standard in recent work in ML-based forecasting.  The method is simple and effective, and this brings attention to a potentially useful tool to have in the forecasting toolbox.


***Update:***
I feel the authors adequately addressed most of my comments, and provided a lot of useful follow-on study and analyses and additional experiments which are much appreciated and really provide a lot more useful information.  Some of it is useful even for providing more info. on the performance of the base models such as varying the input window length to see the impact on the forecast errors.  It would have been good to see even smaller input window lengths however, as I would expect a behavior where at some point if the input window is too small the method is not beneficial, and if the window becomes too big, it should not be much different from the baselines.

Overall, I feel the set of results are impressive and useful.  I agree with the authors that it's useful to bring the approach to the attention of the community (as there are some similar approaches used in some communities but not well known or used in this community), and also the motivation and perspective comes from very different places compared to the similar approaches, and the idea of how it affects the data distribution and showing results along that direction is also new compared to the prior similar approaches.  It provides a widely applicable and effective method, with thorough experiment results and study.  I also feel it could motivate future work in a direction not too well considered before.  Therefore I think it is worthy of acceptance despite the actual algorithmic approach contributed itself being incremental and simple, and given the responses and revision, decide to change my rating.

---

> ### Author Response · Authors · 2021-11-17
> **Response to Reviewer V5fo (Q1)**
>
> We appreciate the thorough review and the positive comments regarding our paper. The main issues raised are the limited novelty of our model, the lack of comparison with the existing hybrid models, and the missing details on the hyperparameter setting. We would like to respond to your comments and hope that we address all your concerns below:
>
> **Q1. My main hesitance to accept the paper is the lack of novelty - as sliding window normalization is already commonly used, and the method is a very minor change. I feel this could possibly be improved with the ablation study comparing the results of the proposed method without the learnable parameters gamma and beta.**
>
> A1.
> Regarding the paper about Adaptive normalization [1], we found it similar to our method in that the normalization method collects statistics from the current input time-series data and then uses the values to normalize and denormalize the data. We added the discussion on this paper in the related work in Section 3.
> However, the paper, **Adaptive norm**, has three main differences from our method as follows, other than the learnable affine transformation that you kindly clarified.
>
> First, the most significant difference is the normalization formula applied to the data, which determines how the data distributions to be changed after the normalization; given a time-series signal $x \in \mathbb{R}^T$ in the current window, where $T$ is the window size, RevIN computes the mean $\mu=\frac{1}{T}\sum_{i=1}^Tx_i$ and the variance $\sigma^2=\frac{1}{T}\sum_{i=1}^{T}(x_i-\mu)^2$, normalizes $x$ as $\frac{x-\mu}{\sigma}$ and then applies the affine transformation with learnable parameters $\gamma$ and $\beta$ as $\gamma\cdot \frac{x-\mu}{\sigma} + \beta$. On the other hand, the Adaptive norm computes only the mean $\mu=\frac{1}{T}\sum_{i=1}^Tx_i$ and normalizes the current data as $\frac{x}{\mu}$.
> Afterward, the Adaptive norm additionally applies min-max normalization to make the values in $[-1, 1]$ using the global minimum and the maximum values computed using the entire data, not using the current sliding window.
>
> These may be viewed as minor differences, but they play a critical role in solving the distribution shift problem. Whereas RevIN makes every instance of a variable (or attribute) have the same mean and variance, the Adaptive norm only scales each instance with the mean; the instances still have different values for the mean and variance, potentially suffering from the distribution shift problem existing in the original data.
> Moreover, the Adaptive norm is based on the assumption of *disjoint* sliding windows on each input sequence, which does not allow the overlap between the sliding windows, leading to limited applicability; it is not compatible with the standard setting of recent deep learning-based approaches, which causes a significant reduction of the number of training data items. In contrast, our method is flexible to experimental settings, having no requirement to use.
> Another important difference lies in the underlying assumption. The Adaptive norm enforces the prediction output to have the identical minimum and maximum values to input, assuming that the model input and output have the same statistics. On the other hand, RevIN has no restrictions on the relation of the input and output distribution. Instead, RevIN allows the model to focus on *capturing the input and output statistics difference* by removing all their common non-stationary statistics (**Appendix A.9**).
> These subtle differences make our approach significantly improve the forecasting performance in multivariate time-series while increasing the applicability of the method.
>
> Additionally, we have a fundamental difference with **DAIN** as well, except for the absence of the built-in denormalization step. DAIN learns to transform the mean and variance values of the current input and use the new values to normalize the data. As a result, the input sequences can have different means and variances from each other, which indicates that the distribution shift can still exist after the normalization. As shown in our paper, the denormalization step shows a critical role in improving the model performance by recovering the prediction to the original value. However, a denormalization step cannot be added to DAIN since it has the Hadamard multiplication operation at the last step, which is not reversible when the input and prediction sequence lengths are different. These differences would be the reason for its inferior performance to RevIN, even requiring more computational costs and a larger amount of model parameters (at least three $D \times D$ matrices, where $D$ is the number of variables).

---

> > ### Author Response · Authors · 2021-11-17
> > **Response to Reviewer V5fo (Q1)**
> >
> > Another main difference between our method and prior work is the learnable affine transformation. Thus, we ablate the affine transformation from RevIN to see its impact on the performance. We conduct the experiments on the six datasets using N-BEATS as the baseline. We report the average and the standard deviation values for the five runs.
> >
> > |                 |       | **w/o affine.**  |           | **w/ affine.**                |(ours)|
> > |-----------------|-------|-------------------------|-------------|-------------------------------|-----------------|
> > |                 | $T_y$ | **MSE**                 | **MAE**     | **MSE**                       | **MAE**         |
> > | **ETTh1**       | **48**    | 0.370±0.003              | 0.393±0.003 | **0.363±0.002**               | **0.389±0.003** |
> > |                 | **960**   | 0.675±0.034             | 0.576±0.014 | **0.638±0.019**               | **0.559±0.017** |
> > | **ETTh2**       | **48**    | 0.257±0.003             | 0.322±0.002 | **0.254±0.008**               | **0.321±0.005** |
> > |                 | **960**   | 0.483±0.012             | 0.487±0.007 | **0.471±0.015**               | **0.481±0.008** |
> > | **ETTm1**       | **96**    | 0.384±0.013             | 0.408±0.009 | **0.378±0.011**               | **0.406±0.007** |
> > |                 | **1344**  | 0.664±0.085             | 0.567±0.039 | **0.631±0.061**               | **0.556±0.020**  |
> > | **ECL**         | **48**    | 0.197±0.002             | 0.302±0.002 | **0.195±0.002**               | **0.301±0.001** |
> > |                 | **960**   | 0.347±0.034             | 0.415±0.026 | **0.325±0.019**               | **0.398±0.015** |
> > | **Air quality** | **48**    | 0.707±0.009             | 0.601±0.002 | **0.705±0.019**               | **0.600±0.0090**   |
> > |                 | **720**   | 0.852±0.025             | 0.689±0.013 | **0.842±0.015**               | **0.686±0.008** |a variety
> > | **Nasdaq**      | **30**    | 0.983±0.017             | 0.565±0.005 | **0.981±0.017**               | **0.564±0.005** |
> > |                 | **60**    | 1.163±0.010             | 0.610±0.002  | **1.155±0.020**                | **0.608±0.005** |
> >
> > The results show that the affine transformation contributes to performance improvement consistently on a variety of datasets.
> >
> > Moreover, the learnable affine transformation in RevIN is another important difference from prior sliding window normalization methods. While existing approaches are a preprocessing & postprocessing method applied outside of the main prediction model, RevIN is an end-to-end trainable layer that can be added to any layer in the model like batch norm [2] and instance norm [3], which are the recently proposed deep learning-based normalization layers.

---

> > > ### Author Response · Authors · 2021-11-17
> > > **Response to Reviewer V5fo (Q1)**
> > >
> > > As mentioned in our paper, a model can add RevIN in an intermediate layer, even to several layers. Thus, we verify that adopting RevIN to the intermediate layers instead of the input and output layers can improve the forecasting performance as well. We add RevIN at the first stack of N-BEATS and SCINet and evaluate their performance on the six datasets. We provide the results on only the ETTh2 dataset due to the lack of space. Please refer to **Table 7 in Appendix A.4** for the entire results.
> > >
> > > |                       |         | Prediction          | length            |                     |                     |                     |                     |
> > > |-----------------------|---------|---------------------|---------------------|---------------------|---------------------|---------------------|---------------------|
> > > |                       |         | **24**              | **48**              | **168**             | **336**             | **720**             | **960**             |
> > > | **N-BEATS**           | **MSE** | 0.403±0.185         | 1.33±0.24           | 7.174±0.449         | 4.859±0.268         | 5.656±1.053         | 6.408±2.039         |
> > > |                       | **MAE** | 0.472±0.101         | 0.918±0.073         | 2.329±0.049         | 1.863±0.043         | 2.012±0.186         | 2.077±0.242         |
> > > | **+RevIN   (inter.)** | **MSE** | 0.199±0.002         | 0.263±0.007         | 0.425±0.015         | **0.446**±**0.007** | 0.505±0.022         | 0.523±0.04          |
> > > |                       | **MAE** | 0.291±0.002         | 0.335±0.005         | 0.434±0.01          | 0.456±0.005         | 0.501±0.013         | 0.522±0.025         |
> > > | **+RevIN   (i/o.)**   | **MSE** | **0.192**±**0.003** | **0.254**±**0.011** | **0.41**±**0.01**   | 0.449±0.011         | **0.496**±**0.008** | **0.471**±**0.015** |
> > > |                       | **MAE** | **0.276**±**0.002** | **0.32**±**0.008**  | **0.418**±**0.005** | **0.447**±**0.006** | **0.482**±**0.002** | **0.481**±**0.008** |
> > > | **SCINet**            | **MSE** | 0.199±0.026         | 0.35±0.025          | 0.559±0.044         | 0.664±0.073         | 1.546±0.378         | 1.862±0.153         |
> > > |                       | **MAE** | 0.295±0.027         | 0.422±0.027         | 0.518±0.025         | 0.583±0.03          | 0.944±0.141         | 1.066±0.055         |
> > > | **+RevIN   (inter.)** | **MSE** | 0.186±0.003         | 0.313±0.045         | 0.338±0.003         | 0.422±0.001         | 0.634±0.01          | 0.734±0.014         |
> > > |                       | **MAE** | 0.272±0.001         | 0.373±0.032         | 0.38±0.001          | 0.443±0.001         | 0.564±0.005         | 0.603±0.005         |
> > > | **+RevIN   (i/o.)**   | **MSE** | **0.18**±**0.004**  | **0.231**±**0.006** | **0.337**±**0.007** | **0.357**±**0.003** | **0.411**±**0.003** | **0.438**±**0.007** |
> > > |                       | **MAE** | **0.263**±**0.002** | **0.302**±**0.006** | **0.378**±**0.003** | **0.403**±**0.002** | **0.445**±**0.002** | **0.462**±**0.004** |
> > >
> > > The above results demonstrate that even when added to the intermediate layers, RevIN improves the performance of the baselines as a learnable normalization layer. We mainly focus on adding RevIN to the input and output of a model since it shows robust performance on average. Nevertheless, the model adopting RevIN in the intermediate layers also shows the best performance frequently, consistently outperforming the baseline without RevIN. This performance is even better than the dynamic normalization methods, LSTNet and ES-RNN, when comparing them where N-BEATS is used as the baseline (please refer to **Table 9 in Appendix A.6**). For example, when the prediction length is 960, the mean squared errors of LSTNet, ESRNN, RevIN (inter.), and RevIN (i/o) are 5.627, 1.338, 0.523, and 0.471, on average.

---

> > > > ### Author Response · Authors · 2021-11-17
> > > > **Response to Reviewer V5fo (Q1)**
> > > >
> > > > In summary, RevIN fundamentally differs from existing sliding window normalization methods, which are preprocessing & postprocessing steps; **RevIN is a flexible, end-to-end trainable layer that can be applied to any arbitrarily chosen layers, effectively suppressing non-stationary information (mean and standard deviation of the instance) from one layer and restoring it on the other layer at a virtually symmetric position. Despite its remarkable performance, there has been no work on generalizing and expanding the instance-wise normalization & denormalization as a flexibly applicable, trainable layer in the time-series domain.**
> > > >
> > > > Recently, deep learning-based time-series forecasting approaches, such as Informer and N-BEATS, have shown outstanding performance in time-series forecasting. However, they overlooked the importance of normalization, merely using simple global preprocessing to the model input without further exploration and expecting their end-to-end deep learning model to replace the role.
> > > > Despite the simplicity of our method, there have been no cases of using such techniques in modern deep-learning-based time-series forecasting approaches, e.g., N-BEATS, Informer, and SCINet, as the reviewer kindly stated.
> > > >
> > > > **In this sense, as our proposed approach has not been aware in the community, our contribution would be enlightening the importance of the appropriate normalization method in deep-learning-based time-series approaches; we propose a carefully designed, deep-learning-friendly module, for time-series forecasting, by combining the learnable affine transformation to the method, which has been widely accepted in recent deep-learning-based normalization work** (Batch norm [2], Instance norm [3]).
> > > >
> > > > In conclusion, we propose a generally-applicable normalization-and-denormalization method with learnable affine transformation, symmetrically structured to remove and restore the statistical information of a time-series instance, and demonstrate its effectiveness through extensive quantitative and qualitative results, addressing the distribution shift problem; it leads to significant performance improvements in the work on time-series forecasting, which will constitute our significant contribution. Please refer to the updated manuscript (mainly the appendix) to see the additional experimental results demonstrating the effectiveness of RevIN.

---

> ### Author Response · Authors · 2021-11-17
> **Response to Reviewer V5fo (Q2)**
>
> **Q2.  Additionally it would also be useful to see the results of a hybrid approach with the baselines as a commonly used alternative to the proposed dynamic normalization approach (e.g., fit a linear AR model + the baseline model).**
>
> A2.
> We compare RevIN with existing hybrid methods, LSTNet [4] and ES-RNN [5]. Similar to adding RevIN to the baseline model, we add the autoregressive linear bypass module of LSTNet and the modified Holt-Winters exponential smoothing of ES-RNN to the baseline model, respectively.
> Using N-BEATS as the baseline model, the experiments are conducted on the four datasets used in our paper and the M4 dataset used in the original paper of N-BEATS.
> Due to the insufficient space, we provide the results on the ETTh2 dataset only, but please refer to **Table 9 and Fig. 8 in Appendix A.6** to see the entire table and the illustration of distributions of the hybrid models.
>
> |       |         |   N-BEATS   |             |  +LSTNet*  |             |  +ES-RNN*  |             |    +RevIN   |             |
> |:-----:|:-------:|:-----------:|:-----------:|:-----------:|:-----------:|:-----------:|:-----------:|:-----------:|:-----------:|
> |       |         |     **MSE**     |     **MAE**     |    **MSE**     |     **MAE**     |     **MSE**     |     **MAE**   |     **MSE**     |     **MAE**    |
> | **ETTh2** |    **24**   | 0.403±0.185 | 0.472±0.101 | 0.394±0.099 | 0.485±0.068 |  0.614±0.01 | 0.522±0.004 | **0.192±0.003** | **0.276±0.002** |
> |       |    **48**   | 1.330±0.240 | 0.918±0.073 | 1.261±0.214 | 0.907±0.075 | 0.654±0.009 | 0.543±0.007 | **0.254±0.011** |  **0.32±0.008** |
> |       |   **168**   | 7.174±0.449 | 2.329±0.049 | 7.053±0.428 | 2.290±0.095 | 0.962±0.129 | 0.696±0.054 |  **0.41±0.010** | **0.418±0.005** |
> |       |   **336**   | 4.859±0.268 | 1.863±0.043 | 5.070±0.336 | 1.914±0.083 | 1.204±0.158 | 0.789±0.050 | **0.449±0.011** | **0.447±0.006** |
> |       |   **720**   | 5.656±1.053 | 2.012±0.186 | 6.311±2.057 | 2.049±0.225 | 1.284±0.145 | 0.810±0.033 | **0.496±0.008** | **0.482±0.002** |
> |       |   **960**   | 6.408±2.039 | 2.077±0.242 | 5.627±1.670 | 1.965±0.314 | 1.338±0.535 | 0.809±0.127 | **0.471±0.015** | **0.481±0.008** |
> |   **M4 (x10)**  | **average** | 0.224±0.004 | 0.207±0.003 | 0.223±0.004 | 0.206±0.003 | 0.223±0.001 | 0.204±0.001 | **0.208±0.001** | **0.197±0.001** |
>
> The autoregressive linear bypass module of LSTNet also consistently reduces the prediction error compared to the baseline, but the performance improvement is smaller than our method. For example, when the prediction length is 960, N-BEATS shows an average error of 6.408, and LSTNet reduces the error to 5.627, but which is still much higher than RevIN that reduces the error to 0.471.
> In the case of ES-RNN, it often degrades the baseline performance, e.g., when the prediction length is 24, but it significantly reduces the error for the long prediction length much better than LSTNet, but still worse than RevIN, e.g., when the prediction length is 960.
> We further analyze the data distributions of the methods in **Fig. 8 in Appendix A.6**.
> The results show that RevIN and ES-RNN make sequences have similar statistics, thereby alleviating the distribution shift problem in the input data. This would lead to competitive performance on the long prediction sequence in contrast to LSTNet. Also, in LSTNet and ES-RNN, the distributions of the final output significantly differ from the original data, which implies that they fail to learn the appropriate data distribution, and this would be the main reason for higher prediction error than RevIN.

---

> ### Author Response · Authors · 2021-11-17
> **Response to Reviewer V5fo (Q3)**
>
> **Q3.  I would also like to see more analyses and discussion around the impact of the window length used, and fixes / additional details around method and experiment details.**
>
> A3.
> We revise the manuscript to include analysis of the impact of the input sequence length on RevIN in **Appendix A.11** and further hyperparameter details in **Appendix A.12**.
>
> Regarding the questions on the window size, we first clarify that the input sequence length is set as equal to the (sliding) window length in our work. We compute the mean and variance across the entire input sequence length and use the statistics at the RevIN layers. We show the effect of the input sequence length on the forecasting performance on the ETTh1 dataset as shown in the following table. We increase the model input sequence length (‘$T_x$’ in the below table) from 48 (two days) to 960 (40 days) to see its impact on RevIN when the prediction length is fixed to 960 (40 days). The average errors for five runs with the standard deviation values are reported in the table. We report only the results of Informer due to the lack of space. Please refer to **Fig. 7 in Appendix A.3** to see the entire results as graphs.
>
> |     |     **N-BEATS** |             |      **+RevIN** |             |
> |:---:|:------------:|:------------:|:------------:|:------------:|
> |  **$T_x$**   |     **MSE**     |     **MAE**    |     **MSE**     |     **MAE**     |
> |  **48** | 0.907±0.026 | 0.730±0.014 | **0.654±0.048** | **0.552±0.022** |
> | **168** | 1.046±0.072 | 0.805±0.032 | **0.701±0.059** | **0.580±0.022** |
> | **336** | 1.495±0.333 | 0.959±0.113 | **0.652±0.013** | **0.567±0.005** |
> | **480** | 1.383±0.033 | 0.932±0.025 | **0.678±0.019** | **0.575±0.009** |
> | **720** | 2.714±1.350 | 1.150±0.125 | **0.755±0.089** | **0.608±0.038** |
> | **960** | 4.161±4.955 | 1.364±0.640 | **0.767±0.082** | **0.626±0.031** |
>
> The results show that RevIN consistently outperforms the baseline for various input sequence lengths. More importantly, RevIN makes the baseline models more robust to the input length. In other words, the proposed method shows stable performance for various input lengths in contrast to the baseline models, which shows a high variance in their performance according to the input length.
> Notably, in N-BEATS, the average error, as well as the standard deviation of the error, significantly increase as prolonging the input length.
> This is because the trend of data is expressed as a linear function in N-BEATS. As the input length becomes prolonged, there is a chance that the variance (or non-stationarity) in the time-series values becomes higher, decreasing the accuracy of the linear trend to fit the data unless the data is monotonically increasing or decreasing. Also, the mispredicted trends linearly increase the error in the future values.
> However, when N-BEATS adopts RevIN, it removes the non-stationary statistics from the input, and thus, the model shows robust performance against the input length.
> Removing the variability, i.e., normalizing its mean and standard deviation, before feeding it to the model and returning it to the output would make the model learning stable.
>
>
> In our paper, we use the same input sequence length for Informer and N-Beats. Similar to Informer, we select the input sequence length from {2 days, four days, seven days, 14 days, 15 days, 20 days, 30 days} for the ETTh1, ETTh2, and ECL dataset, and {half day, one day, 3.5 days, seven days} for ETTm1 dataset. Particularly, we set the ratio of input length to prediction length to be smaller from 2.0 to 0.35 as the prediction length becomes longer. In the case of SCINet, we follow the identical setting of the original paper for the input length since the method requires the input sequence length to meet a specific condition.
> Regarding the other hyperparameters, we exactly follow the experimental settings of the baseline models by using their official code to conduct experiments except for N-BEATS that have no officially released code. We reproduce the model and set hyperparameters as stated in the original N-BEATS paper. Additionally, we conduct a grid search for the learning rate of N-BEATS with the range of [1e-5, 1e-3] and train the model with the weight decay with the factor of 0.001 to stabilize training.

---

> ### Author Response · Authors · 2021-11-17
> **Response to Reviewer V5fo (Q4, Q5)**
>
> **Q4. It would also best to show the results on a larger variety of datasets, even those used in prior work by the base methods - as the benefit might only be for this common type of data used here.**
>
> A4.
> Please see the general response, titled **“General Response: Experimental Results on Additional Real-World Datasets.”**
>
>
>
>
> **Q5.**
> 1) **The mathematical formulation is not precise / changes at different places. E.g., at first instances are referred to as $x^{(i)}$, then suddenly referred to by $x_k$.**
> 2) **Alg. 1 definition directly contradicts the description of the method and definitions in the main paper. (…) However, in Alg. 1 they are said to be in $R^{N \times K}$ - which implies a unique value per window and feature/variable is learned, which doesn't make sense (as then one wouldn't be able to apply it for future forecasts, e.g., series N+1 where we don't have a parameter).**
> 3) **It would be best to make clear the dimensions of the different parameters, the dimensions of the inputs and outputs, and data**
>
>
> A.5
> 1) $x_{kt}$ denotes an observation of the $k$-th variable at a time step $t$, while $x^{(i)}$ indicates the $i$-th sequence (or instance).  That is, $x_{kt}$ indicates $x^{(i)}_{kt}$ in the manuscript. For clarification, we add all the missing notations of $(i)$ in Section 3.1.
> 2) In Alg 1, it is correct that $\gamma, \beta \in \mathbb{R}^K$, not $\mathbb{R}^{N\times K}$. We fixed the mistake in Alg 1 in the manuscript.
> 3) Consider $x^{(i)}$ that denote the $i$-th time-series data in the mini-batch. Let $K$, $T_x$, and $T_y$ denote the number of variables, the input length, and the prediction length. Then, the dimensions of the input, the output, and the learnable parameters $\gamma, \beta$ of RevIN are as written in Section 3.1:
> |                           | notation | dimension                 |
> |:---------------------------|:-----------:|:---------------------------|
> | input                     | $x^{(i)}$ | $\mathbb{R}^{(K,   T_x)}$ |
> | output                    | $y^{(i)}$ | $\mathbb{R}^{(K, T_y)}$   |
> | learnable parameter       | $\gamma$  | $\mathbb{R}^{K}$        |
> | learnable parameter | $\beta$   | $\mathbb{R}^{K}$            |
>
> ---
> [1]Ogasawara et al. "Adaptive normalization: A novel data normalization approach for non-stationary time series." IJCNN 2010.
>
> [2] Ioffe et al. Batch normalization: Accelerating deep network training by reducing internal covariate shift. ICML 2015.
>
> [3] Ulyanov et al. Instance normalization: The missing ingredient for fast stylization. arxiv.
>
> [4]Lai  et al. "Modeling long-and short-term temporal patterns with deep neural networks." SIGIR 2018.
>
> [5] Smyl et al. A hybrid method of exponential smoothing and recurrent neural networks for time series forecasting. International Journal of Forecasting, Volume 36, issue 1.

---

### Official Review · Reviewer_uPmn · 2021-11-08

**Correctness:** 4
**Technical Novelty And Significance:** 2
**Empirical Novelty And Significance:** 3
**Recommendation:** 6
**Confidence:** 3

**Main Review:**

**Strengths:**
- Addressing distribution shift is a relevant task to the ICLR community and of crucial importance to deploy deep-learning based model time-series forecasting;
- Despite being simple, the proposed approach is technically sound, easy to implement, and empirically validated on various datasets, split and prediction windows length;
- The experimental evaluation is thorough and allow one's to evaluate the importance of the type of normalization and the effect de-normalization module.

**Weaknesses:**
- In the recent literature [1], complementary metrics are used to evaluate the shape error (DTW) and temporal error (TDI) separately. Here, the authors only evaluate their method with MSE/MAE metric.

&nbsp;

Small typos:
- introduction of Section 3 : RevIN acronyme used two times in a row;
- page  8: "we the batch normalization" => "we use the batch normalization".

&nbsp;

-----------
[1] V. Le Guen & N. Thome. *Shape and Time Distortion Loss for Training Deep Time Series Forecasting Models*. In NeurIPS 2019.

**Summary Of The Paper:**

This work proposes to use a normalization method to address temporal distribution shift in time-series forecasting. The proposed approach, *RevIN*, consists of two steps: instance normalization on input sequences and "de-normalization" of output sequences by re-using statistics (mean and variance) computed during the normalization step. Experiments are conducted on two time-series datasets (ETT and ECL) with varying splits and prediction windows lengths. Results show that RevIN used on top of deep-learning based methods for time-series forecasting (Informer, N-BEATS and SCINet) improves prediction performances, in particular for long sequence prediction. Finally, they conduct an empirical comparison with other normalization methods, such as batch normalization and min-max normalization, to evaluate the adequateness of instance normalization and of the denormalization step.

**Summary Of The Review:**

-----
**Post-rebuttal comments**
The authors have addressed my concerns and engaged with other reviewers concerns. Their rebuttal includes several complementary analysis and experiments which helps to better validate the quality of the approach. Even though the proposed approach is straightforward, the idea behind RevIN is technically sound and the thorough experimental evaluation is conclusive. Thus, I maintain my rate in favor of an acceptance.
------

While instance normalization is a simple and well-known method in the distribution shift literature, its application for time-series forecasting is sound and empirically validated in this paper. Overall, I tend to vote for accepting but authors should provide complementary metrics raised in weaknesses to further analyze the effect of their approach.

---

> ### Author Response · Authors · 2021-11-17
> **Response to Reviewer uPmn**
>
> We appreciate your review and the positive comments regarding our paper. We would like to respond to your comments as follows, and we hope that we address all your concerns below:
>
> **Q1. In the recent literature [1], complementary metrics are used to evaluate the shape error (DTW) and temporal error (TDI) separately. Here, the authors only evaluate their method with the MSE/MAE metric.**
>
> A1. We evaluate our method using the complementary metrics, DTW and TDI, on all four datasets, ETTh1, ETTh2, ETTm1, and ECL datasets, using the three SOTA models as the baseline.
> We only provide the results on the ETTh2 dataset in the following table due to the lack of space. Please refer to **Table 8  in Appendix A.5** in the revised manuscript to see the entire table.
>
> |              |         | **24**          | **48**          | **168**          | **336**          | **720**            | **960**            |
> |--------------|---------|-----------------|-----------------|------------------|------------------|--------------------|--------------------|
> | **Informer** | **TDI** | 2.016±0.152     | **4.462±0.567**     | **24.016±3.355**     | 74.168±9.015     | 129.44±22.426      | 177.336±18.126     |
> |              | **DTW** | 2.481±0.353     | 8.194±0.397     | 32.496±1.563     | 29.549±2.453     | 35.908±2.230       | 38.348±1.665       |
> | **+RevIN**   | **TDI** | **1.528±0.142** | 4.802±0.342 | 24.516±1.705 | **55.817±3.180** | **111.842±18.634** | **170.218±14.187** |
> |              | **DTW** | **1.601±0.029** | **2.806±0.081** | **6.95±0.313**   | **9.572±0.270**  | **13.109±0.391**   | **15.24±0.246**    |
> | **N-BEATS**  | **TDI** | 1.476±0.230     | 4.244±0.470     | 42.27±1.708      | 95.645±6.432     | 185.321±17.061     | 281.72±75.629      |
> |              | **DTW** | 2.307±0.488     | 6.131±0.542     | 28.487±0.642     | 30.837±1.115     | 47.461±6.122       | 53.148±9.408       |
> | **+RevIN**   | **TDI** | **1.128±0.084** | **2.529±0.165** | **14.759±1.149** | **42.654±3.450** | **92.521±33.126**  | **100.142±29.742** |
> |              | **DTW** | **1.409±0.007** | **2.207±0.010** | **5.242±0.104**  | **8.063±0.155**  | **11.953±0.977**   | **12.766±0.734**   |
> | **SCINet**   | **TDI** | 1.21±0.128      | 3.368±0.380     | 12.307±1.637     | 28.823±2.681     | 142.118±36.524     | 218.477±27.365     |
> |              | **DTW** | 1.35±0.095      | 2.362±0.183     | 4.899±0.229      | 7.447±0.350      | 16.428±2.748       | 20.498±1.979       |
> | **+RevIN**   | **TDI** | **1.088±0.022** | **2.374±0.058** | **10.205±0.533** | **15.831±0.141** | **26.577±0.987**   | **43.028±2.007**   |
> |              | **DTW** | **1.25±0.015**  | **1.906±0.019** | **4.225±0.083**  | **5.975±0.022**  | **8.989±0.035**    | **11.080±0.108**   |
>
>
>
> As shown in the above table, our new experiments show that our approach significantly improves the baseline models across all datasets in terms of the DTW and TDI. Notably, RevIN shows a significant margin compared to the baseline for long prediction length. For example, when RevIN is added, the average DTW decreases from 38.348 to 15.240 for Informer, from 53.148 to 12.766 for N-BEATS, and from 20.498 to 11.080 for SCINet when the prediction length is 960.
> There are few cases where the proposed method predicts a less similar sequence than the baseline, but the margin is minimal in terms of either DTW or TDI compared to the significant margin found when our method outperforms the baseline.
> These results demonstrate that the model adopting RevIN can generate a sequence more similar to the groundtruth than the baseline, especially showing a better prediction accuracy on the long sequence.
>
>
>
>
> **Q2. small typos:**
> * **introduction of Section 3 : RevIN acronyme used two times in a row;**
> * **page 8: "we the batch normalization" => "we use the batch normalization".**
>
> A2.
> All notation errors and typos have been thoroughly revised in the main manuscript and appendix.
>
> ---
> [1] ​​Guen et al. Shape and Time Distortion Loss for Training Deep Time Series Forecasting Models. In NeurIPS 2019.

---

### Author Response · Authors · 2021-11-17
**Summary of Revisions**

**We appreciate all four reviewers for their valuable feedback and positive support.**

**We found our strengths from the reviews as follows:**
1. The proposed approach is straightforward and well-motivated, addressing the distribution shift problem relevant to the ICLR community and crucial for deep-learning-based time-series forecasting work. (Reviewer uPmn, Reviewer V5fo, Reviewer J9jm)
2. The proposed approach demonstrates the widely-applicable benefit of the type of normalization across modern state-of-the-art models and enlights the importance of the appropriate normalization method in deep-learning-based time-series approaches, as our proposed approach has not been aware in the community. (Reviewer V5fo)
3. The proposed approach is technically sound, easy to implement, cheap to compute, and model-agnostic, while the supplementary details and code provided are very helpful for reproducibility. (Reviewer uPmn, Reviewer V5fo, Reviewer J9jm)
4. The experimental evaluation is thorough, and the results are useful, striking, and very extensive, with detailed results in the appendix, including the original reported scores and std. dev. (Reviewer uPmn, Reviewer V5fo)

**Incorporating the reviewer’s feedback, we have uploaded a revised manuscript. A summary of the main changes are as follows:**
- We additionally evaluated our method using the complementary metrics, dynamic time warping (DTW) and temporal distortion index (TDI) in Table 8 in Appendix A.5. (Reviewer uPmn)
- We conducted the ablation study by (1) removing the affine transformation from RevIN and by (2) adding RevIN to the intermediate layers in the model instead of the input and output to verify its effectiveness as a generally-applicable normalization layer to arbitrary deep neural networks. (Reviewer V5fo)
- We compared RevIN with existing hybrid methods, LSTNet and ES-RNN, on the four datasets used in our paper and the M4 competition dataset in Table 9 in Appendix A.6. (Reviewer V5fo)
- We analyzed the impact of the input sequence length on RevIN in Appendix A.1 and added further hyperparameter details in Appendix A.2. (Reviewer V5fo)
- We added the experimental results on three real-world datasets in Table 4 in Appendix A.1 along with qualitative analysis on the Nasdaq dataset to verify the effectiveness of our method on obvious non-stationary time-series. (Reviewer V5fo, Reviewer J9jm)
- We added the theoretical justification regarding the distribution shift problem in Appendix A.9 (Reviewer J9jm)
- All notation errors, typos, and missing details have been thoroughly revised in the main manuscript and appendix. (Reviewer uPmn, Reviewer V5fo, Reviewer J9jm)

**Please refer to the revised manuscript to see the complete results of the new experiments.**

**We hope our responses and revisions address all reviewers’ concerns, and we would greatly appreciate any further comments and clarifications that we can make.**

---

> ### Author Response · Authors · 2021-11-17
> **General Response: Experimental Results on Additional Real-World Datasets.**
>
> **Q4. It would also best to show the results on a larger variety of datasets, even those used in prior work by the base methods - as the benefit might only be for this common type of data used here.” (Reviewer V5fo)**
>
> **Q2. Experiments are on 4 data sets only. To validate the merit of RevIN, I would like to see experimental results on more real-world data sets, for instance those taken from UCI Repository. (Reviewer J9jm)**
>
>
> A. In our paper, we evaluate our method on the four datasets, ETTh1, ETTh2, ETTm1, ECL datasets, which are commonly used in Informer and SCINet original papers.
>
> Additionally, we evaluate our model using three more datasets, including the M4 competition dataset that N-BEATS used in its original paper and two more real-world datasets taken from the UCI repository, the air quality dataset, and the Nasdaq dataset. In total, our proposed method is compared on seven datasets.  We add the experimental results on the additional datasets in **Table 4 in Appendix A.1**, including qualitative results on the Nasdaq dataset and the experimental details. Here, we provide only the results of N-BEATS due to the lack of space.
>
> |             |          |   N-BEATS   |             |    +RevIN   |             |
> |:-----------:|:--------:|:-----------:|:-----------:|:-----------:|:-----------:|
> |             |   **$T_y$**  |     **MSE**     |     **MAE**     |     **MSE**     |     **MAE**     |
> | **Air quality** |    **24**    | 0.698±0.064 | 0.626±0.029 | **0.527±0.005** | **0.498±0.003** |
> |             |    **48**    | 0.955±0.106 |  0.740±0.035 | **0.705±0.019** |  **0.600±0.009**  |
> |             |    **168**   | 1.079±0.108 | 0.818±0.046 | **0.789±0.008** |  **0.660±0.005** |
> |             |    **336**   | 1.105±0.052 | 0.835±0.021 |  **0.860±0.017** | **0.685±0.006** |
> |             |    **720**   | 1.538±0.419 |  0.968±0.110 | **0.842±0.015** | **0.686±0.008** |
> |    **Nasdaq**   |    **30**    |  5.500±0.647  | 1.254±0.086 | **1.023±0.034** | **0.577±0.007** |
> |             |    **60**    | 5.226±0.424 | 1.236±0.032 | **1.207±0.044** | **0.617±0.009** |
> |             |    **120**   | 6.023±0.382 | 1.197±0.034 | **1.959±0.062** | **0.714±0.006** |
> |   **M4 (x10)**  | **average**  | 2.241±0.037 | 2.065±0.029 | **2.082±0.014** | **1.974±0.006** |
>
> As shown in the above table, the model adopting RevIN significantly improves the forecasting performance of the baseline on all three datasets. Notably, RevIN shows outstanding performance on the Nasdaq dataset, reducing the prediction errors by more than half compared to the baseline.
>
> As shown in Fig. 6 in the manuscript, the Nasdaq index (the variable 'Close') has steadily increased since 2002. Accordingly, when data are divided into the train and test data based on a specific point in time, the test data values tend to be higher than the train data values. In other words, the data severely suffers from the distribution shift problem, where the train and test data show a discrepancy in their distribution. As a result, even existing SOTA networks cannot predict the future values appropriately, as shown in Fig. 6. The baseline fails to keep up with the trend of data, whose mean value continues to increase, and thus the prediction results are shifted. However, RevIN mitigates such distribution discrepancy and remarkably increases the prediction performance of the baseline SOTA model. Similarly, RevIN shows superior performance on the decreasing data ('DE1' in Fig. 6), accurately predicting the changing mean of the data.

---

### Decision · Program_Chairs · 2022-01-20

**Decision:**

Accept (Poster)

**Comment:**

This paper introduces the "reversible instance normalization" (RevIN), a method for addressing temporal distribution shift in time-series forecasting. RevIN consists in normalizing (subtracting the mean and dividing by the standard deviation) each layer of of deep neural network in a given temporal window for a given instance, and de-normalizing by introducing learnable shifting and scaling parameters.

The paper initially received one weak accept and two weak reject recommendations. The main limitations pointed out by reviewers relate to the limited novelty of the approach, the positioning with window normalization methods and hybrid methods in times series, and clarifications on experiments. The authors' rebuttal did a good job in answering the main concerns: rV5fo increased its grade from weak reject to clear accept, and RuPmn maintained its weak acceptance recommendation.

The AC carefully read the submission. The AC considers that the idea is simple yet meaningful. The large set of experiments are well conducted and conclusive. The rebuttal successfully answers to relevant issues raised by reviewers, regarding ablation studies (for highlighting the importance of the learnable de-normalization), the impact of the temporal window, the comparison to hybrid approaches and the difference with respect to Adaptive normalization. The AC thus acknowledge that this submission draws important take-home messages for the community, and therefore recommends acceptance.